# Genome-wide meta-analysis identifies 127 open-angle glaucoma loci with consistent effect across ancestries

Puya Gharahkhani [1,190 ✉], Eric Jorgenson [2,190], Pirro Hysi [3,190], Anthony P. Khawaja [4,5,190], Sarah Pendergrass[6,190], Xikun Han [1], Jue Sheng Ong [1], Alex W. Hewitt [7,8], Ayellet V. Segrè[9], John M. Rouhana [9], Andrew R. Hamel[9], Robert P. Igo Jr [10], Helene Choquet [2], Ayub Qassim[11], Navya S. Josyula [12], Jessica N. Cooke Bailey [10,13], Pieter W. M. Bonnemaijer [14,15,16], Adriana Iglesias [14,15,17], Owen M. Siggs [11], Terri L. Young [18], Veronique Vitart [19], Alberta A. H. J. Thiadens [14,15], Juha Karjalainen[20,21,22], Steffen Uebe[23], Ronald B. Melles[24], K. Saidas Nair[25], Robert Luben [5], Mark Simcoe [3,26,27], Nishani Amersinghe[28], Angela J. Cree [29], Rene Hohn[30,31], Alicia Poplawski [32], Li Jia Chen[33], Shi-Song Rong [9,33], Tin Aung[34,35,36], Eranga Nishanthie Vithana [34,37], NEIGHBORHOOD consortium*, ANZRAG consortium*, Biobank Japan project*, FinnGen study*, UK Biobank Eye and Vision Consortium*, GIGA study group*, 23 and Me Research Team*, Gen Tamiya[38,39], Yukihiro Shiga[40], Masayuki Yamamoto [38], Toru Nakazawa[40,41,42,43], Hannah Currant [44], Ewan Birney [44], Xin Wang [45], Adam Auton[45], Michelle K. Lupton[1], Nicholas G. Martin [1], Adeyinka Ashaye[46], Olusola Olawoye [46], Susan E. Williams [47], Stephen Akafo[48], Michele Ramsay [49], Kazuki Hashimoto[40], Yoichiro Kamatani [50,51], Masato Akiyama[50,52], Yukihide Momozawa[53], Paul J. Foster [54,55], Peng T. Khaw [54,55], James E. Morgan[56], Nicholas G. Strouthidis[54,55], Peter Kraft [57], Jae H. Kang [58], Chi Pui Pang[59], Francesca Pasutto [23], Paul Mitchell[60], Andrew J. Lotery [28,29], Aarno Palotie[61,62,63], Cornelia van Duijn[15,64], Jonathan L. Haines [10,13], Chris Hammond [3], Louis R. Pasquale [65], Caroline C. W. Klaver [14,15,66,67], Michael Hauser[68,69,70,71], Chiea Chuen Khor [72], David A. Mackey[7,8,73], Michiaki Kubo[74,191], Ching-Yu Cheng [34,35,36,191], Jamie E. Craig[75,191], Stuart MacGregor [1,191] & Janey L. Wiggs [9,191]

Primary open-angle glaucoma (POAG), is a heritable common cause of blindness world-wide. To identify risk loci, we conduct a large multi-ethnic meta-analysis of genome-wide association studies on a total of 34,179 cases and 349,321 controls, identifying 44 previously unreported risk loci and confirming 83 loci that were previously known. The majority of loci have broadly consistent effects across European, Asian and African ancestries. Cross-ancestry data improve fine-mapping of causal variants for several loci. Integration of multiple lines of genetic evidence support the functional relevance of the identified POAG risk loci and highlight potential contributions of several genes to POAG pathogenesis, including *SVEP1*, *RERE*, *VCAM1*, *ZNF638*, *CLIC5*, *SLC2A12*, *YAP1*, *MXRA5*, and *SMAD6*. Several drug compounds targeting POAG risk genes may be potential glaucoma therapeutic candidates.

A list of author affiliations appears at the end of the paper.

Primary open-angle glaucoma (POAG) is the leading cause of irreversible blindness globally[1,2]. The disease is characterized by progressive optic nerve degeneration that is usually accompanied by elevated intraocular pressure (IOP). Neuroprotective therapies are not available and current treatments are limited to lowering IOP, which can slow disease progression at early disease stages; however, over 50% of glaucoma is not diagnosed until irreversible optic nerve damage has occurred[2,3].

POAG is highly heritable[4,5], and previous genome-wide association studies (GWAS) have identified important loci associated with POAG risk[6–15]. Despite this success, the POAG genetic landscape remains incomplete and identification of additional risk loci is required to further define contributing disease mechanisms that could be targets of preventative therapies.

The majority of known risk loci for POAG have been identified through GWAS in participants of European descent, followed by replication in other ethnic populations. However, previous observational studies have shown that individuals of African ancestry, followed by Latinos and Asians, have higher POAG disease burden compared to those with European ancestry[3,16–18], suggesting important differences in genetic risk and highlighting the need to compare the genetic architecture of these ethnic groups.

In this study, we report the results of a POAG multi-ethnic meta-analysis on 34,179 cases and 349,321 controls, identifying 127 risk loci (44 not previously reported at genome-wide significance levels for POAG). The identified risk loci have broadly consistent effects across European, Asian, and African ancestries. We show that combining GWAS data across ancestries improves fine-mapping of the most likely causal variants for some loci. By integrating multiple lines of genetic evidence we identify the most likely causal genes, some of which might contribute to glaucoma pathogenesis through biological mechanisms related to extracellular matrix cell adhesion, intracellular chloride channels, adipose metabolism, and YAP/HIPPO signaling.

## Results

**Discovery of previously unreported POAG risk loci in Europeans.** We performed a four-stage meta-analysis (Fig. 1). In the first stage, we conducted a fixed-effect meta-analysis of 16,677 POAG cases and 199,580 controls of European descent. The participating studies are detailed in Supplementary Data 1. We identified 66 independent genome-wide significant ($P < 5e-08$) single nucleotide polymorphisms (SNPs) (Supplementary Data 2), of which 16 were not previously identified (i.e., uncorrelated with previously reported SNPs). There was no evidence of inflation due to population structure (linkage disequilibrium (LD) score regression[19] intercept 1.03, se = 0.01, Supplementary Fig. 1A).

**Significant POAG risk loci in Asians and Africans.** In the second stage, we completed a fixed-effect meta-analysis of 6935 POAG cases and 39,588 controls of Asian descent, and a separate fixed-effect meta-analysis of 3281 POAG cases and 2791 controls of African ancestry (Supplementary Data 1). Ten loci were significantly associated with POAG ($P < 5e-8$) in the meta-analysis of Asian studies (Supplementary Data 3), all of which are known POAG loci, and at least nominally ($P < 0.05$) associated with the European meta-analysis. While only one of these loci had a $P < 0.05$ in Africans, eight had consistent direction of effects. For the African meta-analysis, one locus (rs16944405 within *IQGAP1*) reached the genome-wide significance level ($P = 3e-08$). This locus has not been previously reported for POAG, and in this study, was not associated with POAG in Europeans ($P = 0.315$) and Asians ($P = 0.075$) (Supplementary Data 3). The LD score regression intercept was 0.99 (se = 0.009) for Asians and 0.95 (se = 0.006) for Africans, suggesting that these results are not influenced by population structure.

**Consistent genetic effect across ancestries.** As part of the second stage, we replicated the stage 1 European Caucasian findings in an independent dataset comprising 7,286 self-reported cases and 107,362 controls of European descent from the UK Biobank study (UKBB) (Supplementary Data 1)[20], as well as in the meta-analyzed Asian and African datasets described above. We replicated the European Caucasian results in each ancestry group separately (Supplementary Data 2 and Supplementary Fig. 2), followed by combining the three replication datasets in a fixed-

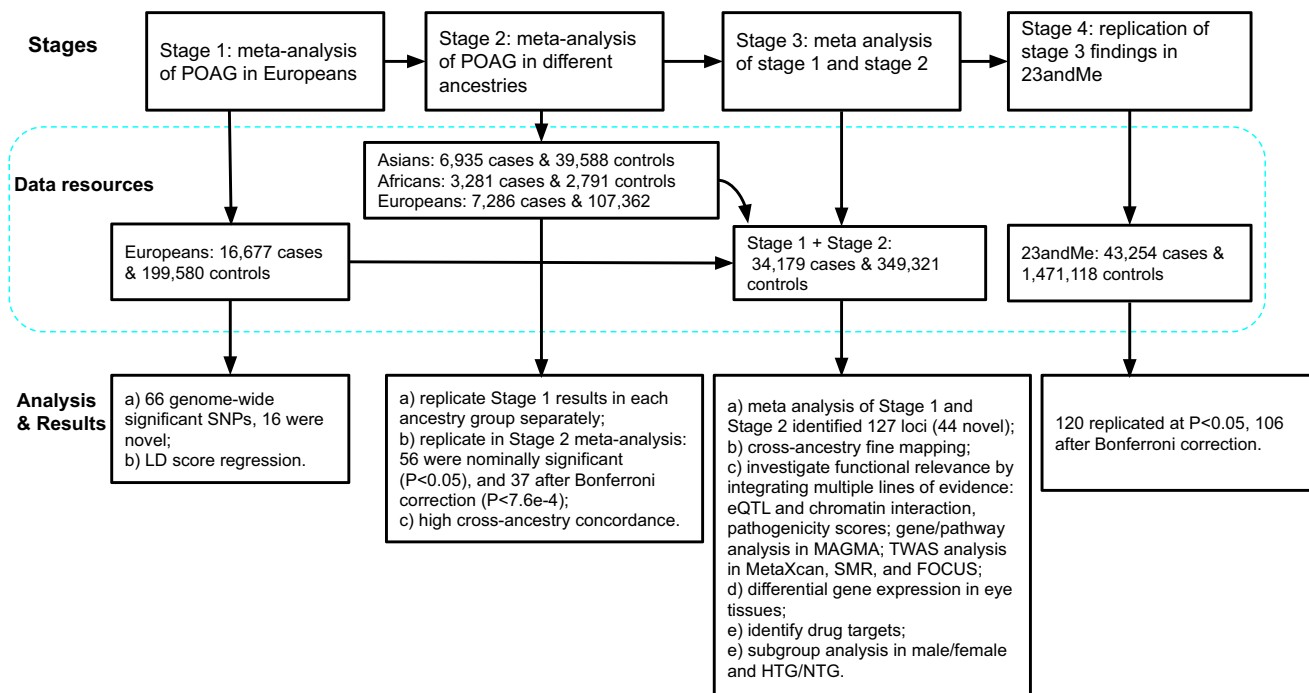

**Fig. 1 Study design.** This figure summarizes the four stages of this study, as well as the data resources and main analyses/results for each stage.

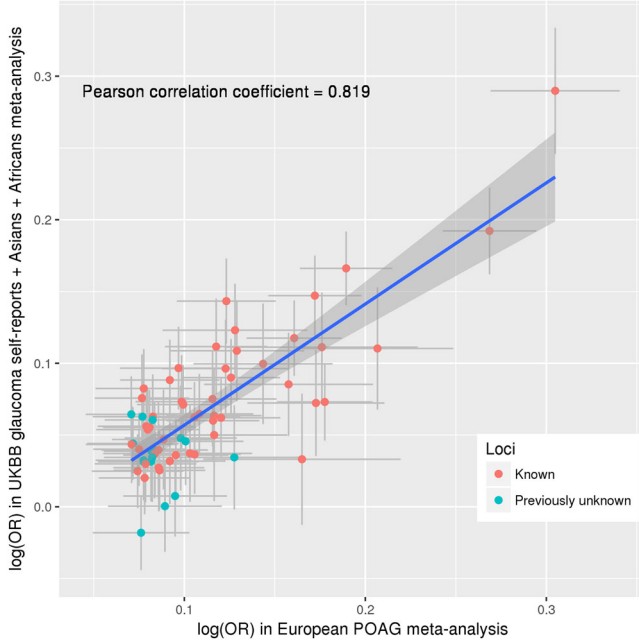

**Fig. 2 Correlation of SNP effect estimates between the European POAG meta-analysis and the replication dataset.** The *x*-axis shows effect estimates in log(OR) scale for the independent genome-wide significant loci obtained from the meta-analysis of POAG in Europeans (16,677 POAG cases vs. 199,580 controls). The *y*-axis shows the effect estimates in log(OR) scale for the same SNPs obtained from meta-analysis of the following three GWAS data: glaucoma self-reports in UKBB, POAG in Asians, and POAG in Africans (the overall sample size of 17,502 cases and 149,741 controls). Red dots are the previously identified risk loci and blue dots are the previously unreported risk loci identified in this study. Horizontal gray bars on each dot represent the 95% confidence intervals (CIs; mean values $+/-1.96*$SEM) for the effect estimates in Europeans (from the GWAS meta-analysis of 16,677 POAG cases vs. 199,580 controls), and vertical gray bars shows the 95% CIs in the replication dataset (from the GWAS meta-analysis of 17,502 POAG cases vs. 149,741 controls). The blue line is the linear regression line best fitting the data. The shaded area shows the 95% CIs on the repression line. UKBB UK Biobank, POAG primary open-angle glaucoma, OR odds ratio.

effect meta-analysis (17,502 cases and 149,741 controls) to maximize statistical power. Of the 66 significant loci in the European Caucasian meta-analysis, 56 were nominally significant ($P < 0.05$), and 37 after Bonferroni correction ($P < 7.6e-04$) in the meta-analyzed replication cohort. The effect sizes had a Pearson correlation coefficient ($r$) = 0.82 (Fig. 2).

There was moderately high cross-ancestry concordance both for genome-wide significant loci and across the genome. For the genome-wide significant SNPs, the European SNP effects were correlated (Pearson correlation coefficient ($r$) = 0.68 [95% confidence intervals (CIs) 0.38–0.97] and $r = 0.44$ [95% CIs 0.20–0.69]) with Asian and African ancestries, respectively (Supplementary Fig. 2B, C). Of the 68 SNPs available in the Asian meta-analysis, 60 (88%) showed the same direction of effect as European Caucasians, and of the 66 SNPs available in the African meta-analysis, 55 (83%) showed the same direction as European Caucasians. The genetic correlation across the genome estimated using the approach implemented in Popcorn v0.9.9[21] was even higher: $r = 0.85$ (95% CIs 0.70–1.00) for European-Asian and $r = 0.75$ (95% CIs −0.93 to 2.43) for European-African. Although the concordance amongst the top SNPs was clear for the European-African comparison, larger sample sizes will be required to narrow the CIs on the European-African genome-wide correlation estimate.

**Discovery of previously unknown POAG risk loci in the cross-ancestry meta-analysis.** In the third stage, given the large genetic correlation between ancestries, we performed a fixed-effect meta-analysis of the results from stage 1 and 2 (34,179 cases vs. 349,321 controls) and identified 127 independent genome-wide significant loci, located at least >1 Mb apart (Fig. 3 and Supplementary Fig. 3). Of these, 44 loci were not previously associated with POAG at genome-wide significance levels (Supplementary Data 4). All loci identified in the European meta-analysis were also significant at the genome-wide level in the combined ancestry meta-analysis except for three loci, two of which were not previously identified (*OVOL2* and *MICAL3*), and one previously reported (*EGLN3/SPTSSA*). Of note, four of the risk loci (*MXRA5-PRKX*, *GPM6B*, *NDP-EFHC2*, and *TDGF1P3-CHRDL1*) are on the X chromosome, representing the first POAG risk loci on a sex chromosome. We also identified an association of a human leukocyte antigen (HLA) gene (*HLA-G/HLA-H*) with POAG. All the lead SNPs have MAF > 0.01 in Europeans, except for two variants: rs74315329 (MAF = 0.0026 in 1000G Europeans) a well-known nonsense variant in *MYOC*[22,23], and rs190157577 (MAF = 0.0013 in 1000G Europeans) an intronic variant in *LINC02141/LOC105371299*.

**The POAG risk loci were strongly replicated in 23andMe.** In the fourth stage, we validated the association of the genome-wide significant SNPs from stage 3 in a dataset comprising 43,254 participants with self-reported POAG (defined as those who reported having glaucoma excluding angle-closure glaucoma or other types of glaucoma) and 1,471,118 controls from 23andMe, Inc. Of the 127 loci, the association results for 125 SNPs were available in 23andMe, 120 of which (96%) were replicated at $P < 0.05$, and 106 (85%) after Bonferroni correction for 125 independent tests (Supplementary Data 4). The correlation of the effect size was $r = 0.98$ (95% CIs 0.977-0.989). In total, the genome-wide significant loci in this study ($N = 127$) collectively explain 9.4% of the POAG familial risk. The previously known loci ($N = 83$) explain 7.5% of the familial risk, and the previously unreported loci ($N = 44$) explain an additional 1.9%.

**Most of the risk loci associated with POAG involve known glaucoma-related endophenotypes.** Several highly heritable endophenotypes are related to POAG risk including IOP, structural variation of the optic nerve characterized as vertical cup-to-disc ratio (VCDR) and variation in thickness of the retina cell layers including the retinal nerve fiber layer (RNFL) and the ganglion cell inner plexiform layer (GCIPL)[24].

All POAG risk variants identified to date have also been associated with either IOP or with VCDR or both. We investigated the association of the POAG loci identified in this study with IOP and VCDR, using previous GWAS data for IOP ($N = 133,492$)[11] and for VCDR ($N = 90,939$)[15]. Figure 4a shows that the majority of loci (89 of 123; four were unavailable for IOP) are also associated with IOP (red and green dots on Fig. 4a). For the 34 loci with unclear effect on IOP (purple and blue dots on Fig. 4a, full data in Supplementary Data 5), we plotted the POAG effect sizes against VCDR effect sizes and determined that 24 of the 32 SNPs (2 SNPs unavailable for VCDR), have a clear effect on VCDR (purple dots on Fig. 4b). Eight of the POAG loci did not appear to have a clear effect on IOP or VCDR, although a small effect on glaucoma via a small change in IOP or VCDR could not be ruled out. The overall correlation of effect sizes between all POAG risk loci and IOP was 0.53, and between POAG and VCDR was 0.31, in line with previously published genetic correlation estimates[25]. To better visualize clustering of the POAG SNPs based on their effect on IOP/VCDR, we created

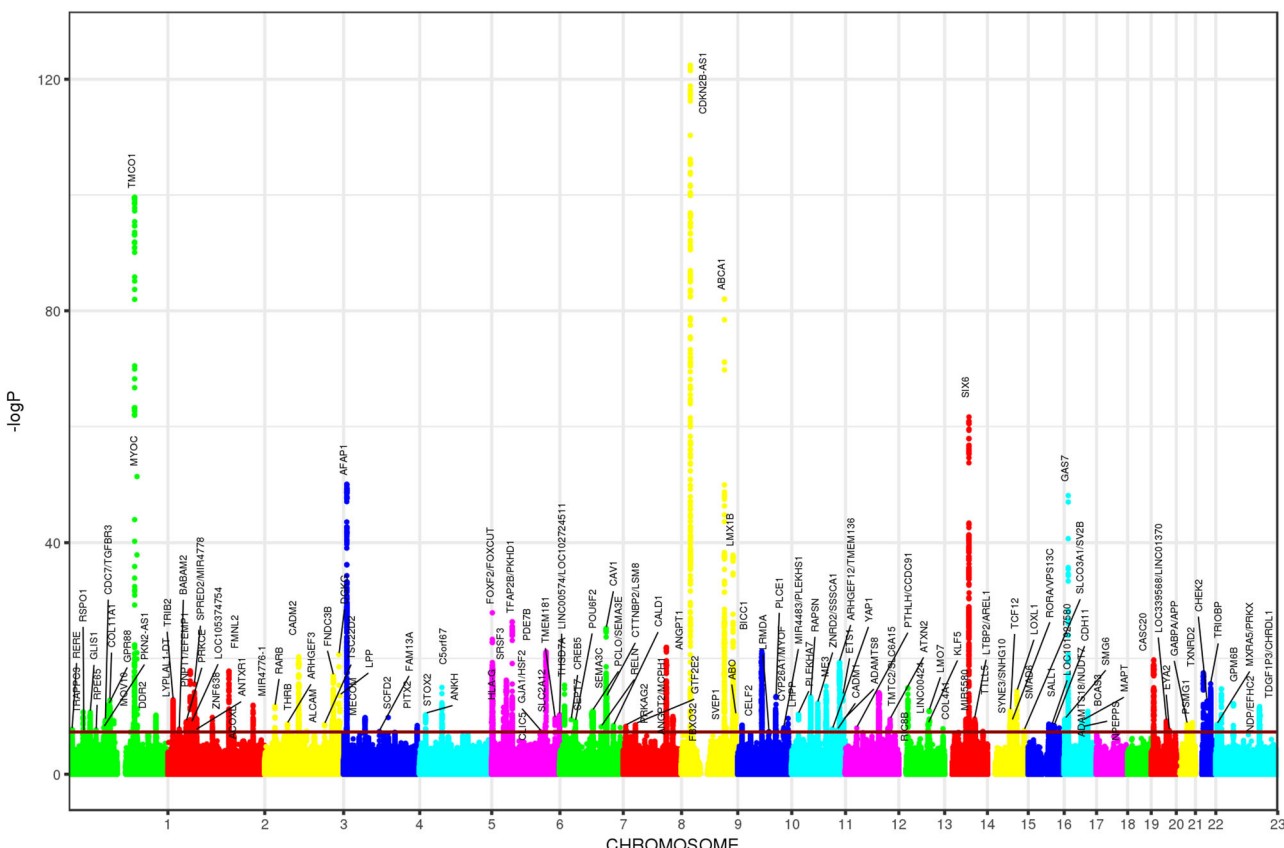

**Fig. 3 Manhattan plots for the cross-ancestry meta-analysis.** Each dot represents a SNP, the x-axis shows the chromosomes where each SNP is located, and the y-axis shows −log10 P-value of the association of each SNP with POAG in the cross-ancestry meta-analysis (34,179 cases vs. 349,321 controls). The red horizontal line shows the genome-wide significant threshold (P-value = 5e-8; −log10 P-value = 7.30). The nearest gene to the most significant SNP in each locus has been labeled.

a heatmap by clustering SNPs based on Pearson correlation between effect estimates of SNPs on POAG, IOP, and VCDR (Supplementary Fig. 4).

Of the 127 POAG genome-wide significant lead SNPs, 116 were available in a GWAS of GCIPL thickness (N = 31,536; Supplementary Data 6). Of these, 14 loci were associated with GCIPL thickness at nominal significance threshold (P < 0.05) and four (PLEKHA7, MAPT, LINC01214-TSC22D2, and POU6F2) after Bonferroni correction for the number of tests (P < 0.05/116). Similarly, 13 loci were nominally associated with RNFL, and three (PLEKHA7, MAPT, and SIX6) after Bonferroni correction. These results suggest that these loci may impact glaucoma pathogenesis through modulation of retinal thickness.

Given that three POAG risk loci that we identified (loci containing MAPT, CADM2, and APP) have also been implicated in Alzheimer's disease (AD) and dementia[26–28], we asked whether the same causal variants may underlie these loci and whether there is evidence for genetic sharing in any of the other genome-wide significant POAG loci. Applying the Bayesian-based colocalization method, eCAVIAR[29], to the cross-ancestry and European POAG GWAS meta-analyses and the publicly available AD GWAS data[30] for 21,982 cases and 41,944 controls[30] of European descent, we found no evidence for sharing of causal variants in the 123 (cross-ancestry) or 66 (European) autosomal POAG loci (Colocalization Posterior Probability (CLPP) < 0.01; Supplementary Data 7). With a larger AD GWAS meta-analysis of 71,880 cases and 383,378 controls[30,31], there was weak support for colocalization at six loci (CLPP = 0.01–0.14; four loci from the cross-ancestry and three from the European POAG meta-analysis

with one overlapping locus; see Supplementary Data 8), though none of these POAG loci reached genome-wide significance in the AD GWAS (AD variant P-values on the order of $10^{-4}$ to 0.05). We note that the colocalization results with this larger AD GWAS meta-analysis[30,31] might be slightly inflated due to the large overlap of UK biobank samples between the POAG and AD meta-analyses. We further estimated the genome-wide genetic correlation between POAG and AD using LD score regression (LDSR)[32] that adjusts for sample overlap, on the two AD GWASs;[30,31] the correlation estimates were 0.03 (95% CIs: −0.11–0.16; P = 0.7) and 0.14 (95% CIs: 0.003–0.28; P = 0.049), respectively.

We next investigated genetic correlation between POAG and a range of other traits using bivariate LDSR[32] through the LD Hub platform (http://ldsc.broadinstitute.org/ldhub/). Only glaucoma, self-report glaucoma, and "Other eye problems" were significantly associated after adjustment for multiple testing for 758 traits (P < 6.6e-05; Supplementary Data 9). Some other traits in UKBB such as myopia (short-sightedness); systolic blood pressure; seeing a psychiatrist for nerves, anxiety, tension or depression; and suffering from nerves, showed some evidence for association at P < 0.003 (Supplementary Data 9).

**Cross-ancestry fine-mapping.** Incorporating GWAS data across European, Asian, and African ancestries allowed us to improve fine-mapping of the most likely causal variants. For 10 loci (including previously unidentified loci GJA1/HSF2, SEPT7, and MXRA5/PRKX), the posterior probability of finding a causal SNP

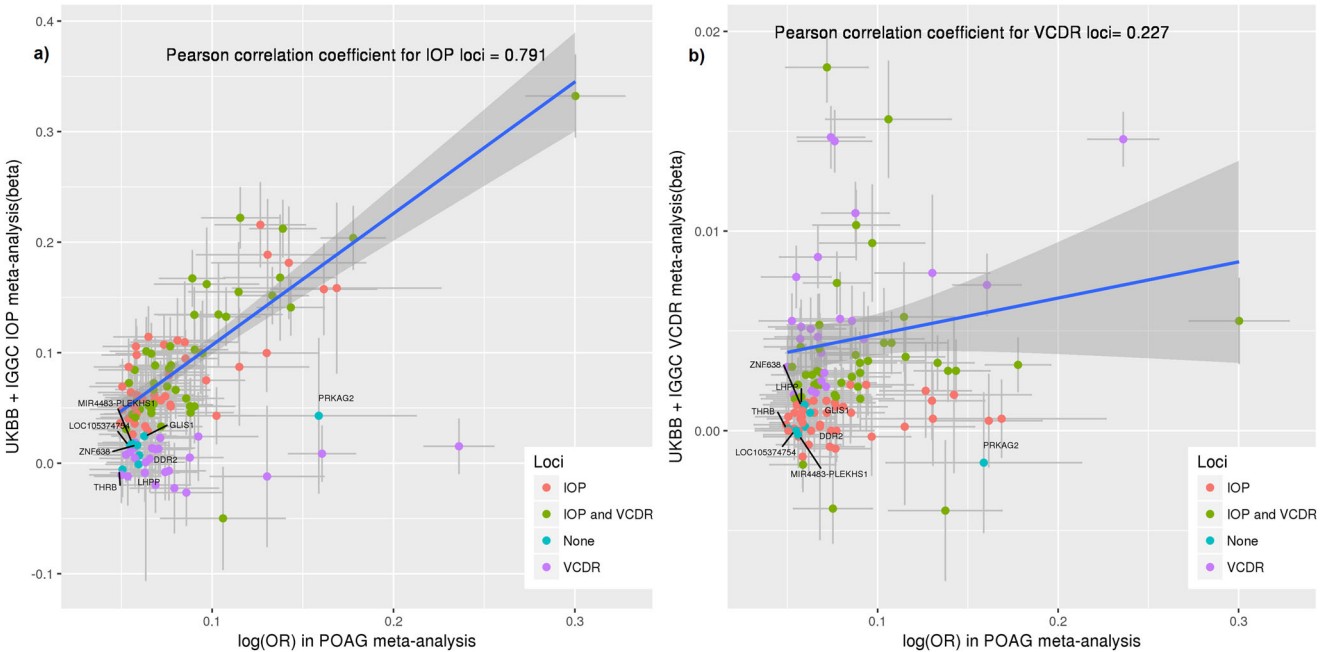

**Fig. 4 Association of the POAG risk loci with IOP and VCDR.** The *x*-axes show POAG effect estimates in log(OR) scale for the independent genome-wide significant loci obtained from the cross-ancestry meta-analysis. The *y*-axes show the effect estimates for the same SNPs obtained from the meta-analysis of IOP in UKBB + IGGC (mmHg scale; **a**) and the meta-analysis of VCDR in UKBB + IGGC (**b**). Blue line shows the regression line for IOP (**a**) and VCDR loci (**b**). Orange dots represent SNPs having *P* < 0.05 for IOP, purple dots *P* < 0.05 for VCDR, green dots *P* < 0.05 for both IOP and VCDR, and blue dots *P* > 0.05 for both IOP and VCDR. Horizontal gray bars on each dot represent the 95% confidence intervals (CIs; mean values + / − 1.96*SEM) for the POAG effect estimates (34,179 cases vs. 349,321 controls), and vertical gray bars shows the 95% CIs for IOP (*N* = 133,492; **a**) and VCDR (*N* = 90,939; **b**). The shaded area shows the 95% CIs on the repression line. Although none of the blue dots show an expected trend of association with IOP in **a** (their 95% CIs do not overlap with the regression line), the majority of them show a trend of association for VCDR in **b**. UKBB UK Biobank, IGGC International Glaucoma Genetics Consortium, IOP intraocular pressure, VCDR vertical cup-to-disc ratio, POAG primary open-angle glaucoma, OR odds ratio.

in Europeans improved after including Asian and African data (improvements from posterior probabilities <0.9 to >0.9 or from <0.8 to >0.8; Supplementary Data 10). For eight loci (of which *THRB* and *SMAD6* were not known), although the posterior probability of a SNP being causal in Europeans was high (>0.9 at least for one SNP), there was still a slight improvement after including the other ancestries (Supplementary Data 10). In contrast, the cross-ancestry data made fine-mapping worse for three loci where the posterior probabilities in Europeans were >0.8 but declined to <0.8 after incorporating data from the other ancestries. For the rest of the loci, the posterior probabilities did not change significantly after including Asian and African data. Overall, the best causal SNPs in Europeans changed for 52 of 127 loci after including data from the other ancestries (Supplementary Data 10). For the remaining 75 loci, at least one SNP remained the best causal SNP in both fine-mapping using European data alone, as well as across ancestries, and 23 of the 127 lead SNPs identified in the meta-analysis remained the best causal SNP in cross-ancestry fine-mapping.

**Gene-based and pathway-based results.** We performed gene-based and pathway-based tests using MAGMA v1.07b[33] for each ancestry separately, followed by combining *P*-values across ancestries using Fisher's combined probability test[34]. We identified 205 genes that passed the gene-based Bonferroni-corrected threshold (*P* < 0.05/20174), corresponding to an additional seven independent risk loci located at least >1 Mb apart from the risk loci identified in the single-variant-based test. Supplementary Data 11 presents significant genes within these seven loci. Expression of the risk genes identified in MAGMA gene-based analysis were significantly enriched in artery and nerve tissues,

reflecting the widely recognized neuronal and vascular character of glaucoma (Supplementary Fig. 5A, B).

Pathway analysis identified 21 significant gene-sets surviving the Bonferroni-corrected threshold (*P* < 0.05/10678). These included previously identified pathways such as collagen formation and vascular development[10,11], and highlighted additional pathways involved in lipid binding and transportation such as apolipoprotein binding and negative regulation of lipid storage (Supplementary Data 12). Genes involved in these pathways that demonstrated suggestive association (*P* < 5e-05) with POAG in the MAGMA gene-based test are summarized in Supplementary Data 13.

**Functional relevance of the identified POAG risk loci.** We used multiple lines of genetic evidence to investigate the functional relevance of the identified risk loci, and to prioritize causal variants and target genes. A summary of these results for the previously unknown loci is provided in Table 1, with additional details presented in Supplementary Data 14. The following paragraphs describe these findings in further detail.

First, the relevance of the identified risk loci was investigated by examining their roles in regulation of gene expression, as well as chromatin interactions. Approximately 76% (96 out of 127) of the lead SNPs or those in high LD (*r*2 > 0.8) with the lead SNPs have also been reported to be significant expression quantitative trait loci (eQTLs) (FDR < 0.05) in various tissues (Supplementary Data 14 and 15). Moreover, the identified risk loci have 34,724 unique significant (FDR < 1e-6) chromatin interactions in various tissues/cell lines involving 4882 genes (Supplementary Fig. 6). Of these, 425 genes overlap with the eQTL genes (286 genes if only considering eQTL results for the genome-wide significant SNPs).

**Table 1 Summary of the previously unreported POAG risk loci annotation and functional studies.**

| Lead SNP | Annotation | Nearest gene | Causal[a] | Chromatin interaction[b] | eQTL any tissue[c] | eQTL retina[d] | Protein altering[e] | CADD >12.37[f] | Gene-based tests[g] | Genes within significant pathways[h] |
|---|---|---|---|---|---|---|---|---|---|---|
| rs172531 | Intron | RERE | NO | RPL7P11 | RERE | RERE | NO | NO | RERE | NO |
| rs941125 | Intron | GLIS1 | NO | SLC25A3P1 | GLIS1 | NO | NO | NO | GLIS1 | NO |
| rs4076000 | Intergenic | ELOCP18/RPE65 | NO | CTBP2P8 | WLS | NO | NO | NO | NO | NO |
| rs12566440 | Intron | GPR88 | NO | VCAM1 | VCAM1 | VCAM1 | NO | NO | GPR88 | NO |
| rs4542196 | Intron | DDR2 | NO | DDR2 | DDR2 | NO | NO | NO | NO | DDR2 |
| rs12623251 | Intergenic | TRIB2 | YES | AC013471.1 | TRIB2 | NO | NO | NO | TRIB2 | NO |
| rs6713914 | Intron | LOC105374754 | NO | AC009970.1 | NO | NO | NO | YES | NO | NO |
| rs12613800 | Intron | ZNF638 | NO | RNU6-105P | ZNF638 | ZNF638 | ZNF638 | NO | ZNF638 | NO |
| rs9852634 | Intron | THRB | YES | RPL31P20 | NO | NO | NO | NO | NO | NO |
| rs1500708 | Intron | ARHGEF3 | YES | ARHGEF3-AS1 | NO | NO | NO | NO | ARHGEF3-AS1 | NO |
| rs6437582 | Intergenic | ALCAM | YES | AC074043.1 | ALCAM | ALCAM | NO | NO | NO | NO |
| rs10517281 | Intron | SCFD2 | YES | Multiple[i] | FIP1L1 | NO | NO | YES | SCFD2 | NO |
| rs57400569 | Intron | FAM13A | YES | Multiple | FAM13A | NO | NO | YES | NO | NO |
| rs17527016 | Intergenic | PITX2 | NO | AC083795.1 | NO | NO | NO | YES | NO | PITX2 |
| rs6552711 | Intron | STOX2 | NO | ENPP6 | NO | NO | NO | NO | NO | NO |
| rs407238 | Intergenic | HLA-G | NO | Multiple | NO | NO | NO | NO | Multiple | NO |
| rs3777588 | Intron | CLIC5 | NO | MIR4642 | CLIC5 | CLIC5 | NO | YES | CLIC5 | NO |
| rs7760346 | Intergenic | GJA1/HSF2 | NO | Multiple | HSF2 | Multiple | NO | NO | NO | GJA1 |
| rs2811688 | Intron | SLC2A12 | NO | TBPL1 | SLC2A12 | SLC2A12 | NO | NO | SLC2A12 | NO |
| rs2191828 | Intron | CREB5 | NO | AC005105.1 | NO | NO | NO | NO | NO | NO |
| rs6957752 | Intergenic | SEPT7 | NO | Multiple | RP11-379H18.1 | RP11-379H18.1 | NO | NO | NO | NO |
| rs7805468 | Intergenic | PCLO/SEMA3E | NO | AC079799.1 | NO | NO | NO | NO | SEMA3E | SEMA3E |
| rs2515437 | Intron | ANGPT2/MCPH1 | NO | Multiple | NO | NO | ANGPT2 | NO | NO | ANGPT2 |
| rs35740987 | Intron | GTF2E2 | NO | Multiple | GTF2E2 | NO | NO | NO | GTF2E2 | NO |
| rs61751937 | Missense | SVEP1 | NO | AL162414.1 | NO | NO | SVEP1 | YES | NO | NO |
| rs6602453 | Intron | CELF2 | NO | AL136369.1 | SFTA1P | NO | NO | NO | ECHDC3 | NO |
| rs72837408 | Downstream | MIR4483/PLEKHS1 | NO | ADRB1 | RP11-211N11.5 | NO | NO | NO | Multiple | NO |
| rs7120067 | Intron | YAP1 | NO | AP000942.2 | YAP1 | NO | NO | NO | YAP1 | YAP1 |
| rs10444329 | Downstream | CADM1 | NO | AP000462.3 | CADM1 | NO | NO | NO | NO | NO |
| rs7972874 | Intergenic | PTHLH/CCDC91 | NO | PTHLH | CCDC91 | CCDC91 | NO | NO | NO | NO |
| rs4903352 | Intron | TTLL5 | NO | TGFB3-AS1 | FLVCR2 | NO | NO | NO | Multiple | TGFB3 |
| rs72692789 | Intron | SYNE3/SNHG10 | NO | Multiple | NO | NO | NO | NO | NO | NO |
| rs2439386 | Intron | SMAD6 | NO | NO | SMAD6 | NO | NO | NO | SMAD6 | SMAD6 |
| rs8038628 | Intergenic | SLCO3A1/SV2B | NO | SLCO3A1 | NO | NO | NO | NO | NO | NO |
| rs190157577 | Intron | LINC02141/LOC105371299 | NO | NO | NO | NO | NO | YES | NO | NO |
| rs242559 | Intron | MAPT | NO | Multiple | Multiple | Multiple | Multiple | NO | Multiple | MAPT |
| rs1348518145 | Intron | NPEPPS | NO | KPNB1 | NO | NO | NPEPPS/TBKBP1 | NO | NPEPPS | NO |
| rs6124885 | Intron | EYA2 | NO | MIR3616 | NO | NO | NO | NO | NO | NO |
| rs13049669 | Intergenic | GABPA/APP | NO | APP | MRPL39 | NO | NO | NO | NO | APP |
| rs13050568 | Intron | PSMG1/LOC107985484 | YES | RPSAP64 | AF064858.11 | NO | NO | NO | NO | NO |
| rs12846405 | Intergenic | MXRA5/PRKX | NO | MXRA5 | MXRA5 | NO | NO | NO | MXRA5 | Multiple |
| rs66819623 | Intron | GPM6B | NO | AC003035.1 | GEMIN8 | NO | NO | NO | NO | NO |
| rs17146835 | Intergenic | NDP/EFHC2 | NO | AL034370.1 | NO | NO | NO | NO | NO | Multiple |
| rs12013156 | Intergenic | TDGF1P3/CHRDL1 | NO | NO | NO | NO | NO | NO | NO | CHRDL1 |

[a]This column indicates whether the GWAS lead SNP was identified to be the most probable causal SNP across ancestries in the multi-ethnic fine-mapping analyses performed using the software PAINTOR.
[b]Where the GWAS lead SNP or SNPs in LD $r^2 > 0.8$ with the lead SNP overlap with one end of a chromatin interaction, the most significant gene involved in this interaction has been shown.
[c]Where the GWAS lead SNP or SNPs in LD $r^2 > 0.8$ with the lead SNP is an eQTL in any GTEx tissue or the other eQTL databases used in this study (see the Methods), the most significant target gene has been shown.
[d]Where the GWAS lead SNP or SNPs in LD $r^2 > 0.8$ with the lead SNP is an eQLT in retina in EyeGEx study, the most significant target gene has been shown.
[e]Where the GWAS lead SNP is a protein-altering variant or in LD $r^2 > 0.8$ with a protein-altering variant, the corresponding gene has been shown.
[f]This column indicates whether the GWAS lead SNP has a CADD score > 12.37.
[g]This column shows the loci in which one or multiple genes were significant in any of the gene-based tests (MAGMA, MetaXcan, SMR, and FOCUS) used in this study. "Multiple" indicates that several genes are involved. These genes have been named in Supplementary Data 14.
[h]This column shows the loci for which the reported nearest genes or the significant genes identified in the gene-based tests above are members of at least one significant POAG pathway identified in this study.
[i]"Multiple" indicates several genes.

In addition, the lead SNPs or SNPs with LD $r^2 > 0.8$ with the lead SNPs for 124 risk loci identified in this study overlap with one end of these chromatin interactions (Supplementary Data 14).

Additional support for the pathogenicity of the POAG risk loci comes from the predicted pathogenicity scores: 20 SNPs had CADD scores >12.37, suggesting that these SNPs have deleterious effects (Supplementary Data 16)[35]. Overall, three lead SNPs are protein-altering variants and 12 lead SNP are in high LD ($r^2 > 0.8$) with a protein-altering variant (Supplementary Data 14), suggesting pathogenic effects through protein-coding roles of these variants (e.g., rs61751937 a missense variant in *SVEP1*). In addition, 24 SNPs had RegulomeDB[36] scores ≤3, supporting regulatory roles for these SNPs (Supplementary Data 16).

To investigate which risk loci are more likely to affect POAG by modulating gene expression, we used two transcriptome-wide association study (TWAS) approaches: Summary Mendelian randomization (SMR)[37] and MetaXcan[38]. For MetaXcan, we used our POAG cross-ancestry meta-analysis statistics, RNA-seq and genotype data from peripheral retina (EyeGEx)[39] and 44 GTEX tissues. Following Bonferroni correction for the maximum number of genes tested ($N = 7209$) in 45 tissues ($P < 1.5e-07$), we identified 100 significant genes, which were selected as the most likely causal genes based on the integration of eQTL data (also see below and Supplementary Data 14 and 17). Of these significant genes, three (*AKR1A1*, *DDIT4L/LAMTOR3*, and *C4orf29*) were located >1 Mb apart from (the other loci) identified using single-

variant and other gene-based tests performed in this study and were not previously reported for POAG (Supplementary Data 17). In a post hoc analysis looking solely at retina, two additional genes (*CNTF* and *MPHOSPH9*) were significant (given Bonferroni correction threshold for 6508 genes in retinal tissue).

Additionally, we integrated our GWAS meta-analysis summary statistics with eQTL data from blood (CAGE eQTL summary data, $N = 2765$) and retina (EyeGEx eQTL data, $N = 406$) using SMR. Given that these eQTL data were obtained from people of European descent, we restricted this analysis to our European meta-analysis to ensure that different gene expression and LD structure patterns between ancestries did not influence the SMR findings. In retina and blood, 16 genes passed the SMR significance threshold corrected for the maximum number of 8516 genes tested in two tissues ($P < 2.9e-06$), of which eight had a $P > 0.05$ in the heterogeneity in dependent instruments (HEIDI) test[37] implemented in SMR, suggesting that the same association signals drive both gene expression and POAG risk, at these loci (Supplementary Data 18). Although the majority of the risk loci identified through the MetaXcan and SMR approaches were also identified in the meta-analysis, these analyses help with prioritizing the most likely functionally relevant genes. To further identify the most plausible causal genes based on gene expression data, we used the approach implemented in FOCUS v0.5[40], a probabilistic framework that assigns a posterior probability for each gene causally driving TWAS associations in multiple tissues (Supplementary Data 19 summarizes the genes with a posterior probability >0.6).

Integrating data from several lines of evidence described above, as well as the cross-ancestry fine-mapping and genetic pathways, provided support for specific genes potentially influencing POAG risk particularly *RERE*, *VCAM1*, *ZNF638*, *CLIC5*, *SLC2A12*, *YAP1*, *MXRA5*, and *SMAD6* (Table 1 and Supplementary Data 14). For example, rs3777588, a lead GWAS SNP in this study, is an intronic variant within *CLIC5*. The CADD score for this variant is 16.58, providing support for the pathogenicity of this variant (determined by a CADD score > 12.37). The lead SNP is also an eQLTL for *CLIC5* in both GTEx and retina, and gene-based analysis by incorporating eQTL data supported the involvement of *CLIC5* in POAG risk. This approach also helped to select best genes near the lead SNPs or to shift the focus from the nearest genes to genes further away. For example, rs12846405, a lead GWAS SNP in this study, is an intergenic variant located between *MXRA5* and *PRKX*. Based on integration of eQTL data and gene-based analysis, *MXRA5* was prioritized as the most likely causal gene in this locus.

**Differential expression of the previously unknown risk loci in eye tissues**. We investigated the expression of the genes nearest to the lead SNP for the novel loci in 21 healthy eye tissues[11] (Supplementary Data 20 and Supplementary Fig. 7). Clustering analysis shows that the majority of these genes were expressed in eye tissues (Supplementary Fig. 7A). We examined the differential expression of the previously unreported genes in ocular tissues likely to be involved in POAG pathogenesis, namely trabecular meshwork, ciliary body and optic nerve head, and found 36/51 (71%) of the genes differentially expressed in these tissues compared to the other eye tissues tested in this study (Supplementary Data 20 and Supplementary Fig. 7B).

**Drug targets**. At least 16 of the POAG risk genes (nearest to the lead SNPs) are targeted by existing drugs, some of which are already in use/clinical trials for several eye or systemic diseases (Supplementary Data 21). The functional relevance of 14 of these 16 drug target genes is supported by the bioinformatic functional

analyses we used in this study (i.e., eQTL, chromatin interaction, etc; Supplementary Data 21). We discuss the relevance of some of these drugs in the discussion section below.

**Sex-stratified meta-analysis**. We identified a very high genetic correlation ($rg = 0.99$, $se = 0.06$) between POAG in men versus women (European stage 1 and UKBB self-reports combined). We also performed cross-ancestry, sex-stratified meta-analyses using a subset of the overall study with sex-stratified GWAS available (Supplementary Data 1; Supplementary Data 22 and 23; Supplementary Fig. 1C, D). Only one signal near *DNAH6* appeared to have a female-specific effect (2:84828363[CA], OR = 1.6, $P = 3.28e-09$ for women; OR = 1.05, $P = 0.56$ for men).

**Subtype-stratified meta-analysis**. Based on IOP levels, POAG can be classified into two major subtypes: high-tension glaucoma (HTG) in which IOP is increased (>21 mmHg), and normal tension glaucoma (NTG) in which IOP remains within the normal range. We performed cross-ancestry subtype-specific meta-analyses using 3247 cases and 47,997 controls for NTG (Normal tension glaucoma defined as glaucoma with IOP < 21 mmHg), and 5144 cases and 47,997 controls for HTG (high-tension glaucoma with IOP > 21 mmHg (Supplementary Data 24 and 25 and Supplementary Fig. 1E, F). All NTG and HTG loci were also significantly associated with the overall POAG meta-analysis except for one locus near *FLNB* that was significant for NTG (lead SNP rs12494328[A], OR = 1.18, $P = 1.7e-08$), but did not reach the significance threshold for POAG overall ($P = 7.5e-07$). However, this SNP was significant in the 23andMe replication study (rs12494328[A], OR = 1.06, $P = 1.35-e12$), and has previously been associated with optic nerve head changes[41]. Overall, all NTG loci were at least nominally associated ($P < 0.05$) with HTG (and vice versa) except for rs1812974 (top SNP near *ARHGEF12*). Although this SNP had the same direction of effect for NTG, the effect was significantly larger for HTG than NTG ($P = 0.007$). Similarly, several other loci had significantly larger effects on one subtype (e.g., *CDKN2B-AS1*, *FLNB*, and *C14orf39* had larger effects on NTG than HTG) (Supplementary Data 24 and 25). Overall, the genetic correlation between NTG and HTG was estimated to be 0.58 ($se = 0.08$) using LD score regression and the meta-analysis summary data from Europeans.

## Discussion

In this large multi-ethnic meta-analysis for POAG, we identified 127 risk loci for POAG, of which 44 were not previously identified. We also identified additional risk loci using gene-based tests and highlighted genetic pathways involved in the pathogenesis of POAG. We observed relatively consistent genetic effects for POAG across ancestries. The risk loci include genes that are highly expressed in relevant eye tissues, nerves, arteries, as well as tissues enriched with these components. Functional relevance of the identified risk loci were further supported by eQTL and chromatin interaction data.

We identified a significant correlation between the POAG effect sizes of genome-wide significant SNPs, as well as all the SNPs throughout the genome, across Europeans, Asians, and Africans. Although previous studies have suggested that the genetic architecture of POAG might differ between Africans and Europeans[42], we observed a moderate correlation ($r \sim 0.45$) between effect sizes of the POAG risk loci in Europeans and Africans (Supplementary Fig. 2C), and the correlation was higher between Europeans and Asians ($r \sim 0.7$). Although the overall correlation is moderately high across ancestries, there are genomic regions where the LD pattern differs by ancestry and our fine-mapping approach showed that incorporating GWAS data

across ancestries improved the probability of finding a causal variant for 18 loci in this study, including known (e.g., *AFAP1* and *RELN* loci) and unknown (e.g., *GJA1/HSF2* and *SEPT7*) loci. However, the most probable causal variants in Europeans remained the same for ~60% (75 out of 127) of the risk loci even after incorporating Asian and African GWASs. Overall, due to the relatively lower statistical power of our African studies, the fine-mapping results in this study were not strongly influenced by African GWASs, emphasizing that larger African POAG GWASs are required for better cross-ancestry fine-mapping in the future.

We also identified a previously unknown association of a human leukocyte antigen (*HLA-G/HLA-H*) with POAG. The HLA system is a gene complex encoding the major histocompatibility complex proteins in humans. These cell-surface proteins are responsible for the regulation of the human immune system. The most significant SNP in this region (rs407238) has been associated at the genome-wide significance level for other traits such as Celiac disease, intestinal malabsorption, disorders of iron metabolism, multiple blood traits, hyperthyroidism, multiple sclerosis, hip circumference, and weight (https://genetics.opentargets.org/variant/6_29839124_C_G). The mechanism of action of the lead SNP appears to be via IOP (UKBB IOP GWAS $P = 8.8e-06$).

The gene-sets enriched for the risk loci identified in this study indicate two major pathogenic mechanisms for POAG: (1) vascular system defects, mainly the molecular mechanisms contributing to blood vessel morphogenesis, vasculature development, and regulation of endothelial cell proliferation, and; (2) lipid binding and transportation—mainly the molecular mechanisms involved in intracellular lipid transport, apolipoprotein binding, negative regulation of lipid storage, and positive regulation of cholesterol efflux. Involvement of the vascular system in the pathogenesis of POAG is further supported by our results showing enrichment of the expression of the POAG risk genes in arteries and vessels. Molecular targets in these pathways can be potential candidates for treatment of POAG. Two of the significant gene-sets in this study (phagocytosis engulfment and negative regulation of macrophage derived foam cell differentiation) suggest an important role of immune system defects in increasing the risk of POAG.

Integrating several lines of genetic evidence provided support for specific genes within the identified risk loci that could influence risk through known and unknown processes. *MXRA5* and *SMAD6* are both involved in transforming growth factor (TGF) beta-mediated extracellular matrix remodeling[43,44], a process known to contribute to POAG risk[45]. Additionally, a *SVEP1* missense allele was associated with POAG risk (rs61751937). *SVEP1* encodes an extracellular matrix protein that is essential for lymphangiogenesis in mice, through interaction with ANGPT2 (the product of another POAG risk gene identified in this study), and modulation of expression of *TEK* and *FOXC2* in knockout mice[46]. Lyphangiogenesis has an important role in the development of Schlemm's canal required for outflow of fluid from the eye[47,48], and two other genes necessary for lyphangiogenesis and Schlemm's canal development (*TEK*, *ANGPT1*) cause childhood glaucoma[49,50]. Interestingly, *SVEP1* was shown to be a modifier of *TEK*-related primary congenital glaucoma[51]. VCAM1 is an extracellular matrix cell adhesion molecule involved in angiogenesis and possibly regulation of fluid flow from the eye[52]. *RERE* mutations are a cause of neurodevelopmental disorders that can involve the eye[53], providing further evidence for a role of ocular development in adult glaucoma[54]. While *RERE* has also been associated with VCDR[55], this is the first association with POAG.

Genes involved in biological processes not previously known to contribute to glaucoma have also been implicated by this study. *CLIC5* encodes a chloride channel that functions in

mitochondria[56] and could have a role in ocular fluid dynamics. ZNF638 is a zinc finger protein that regulates adipose differentiation[57] and has been implicated in the genetic regulation of height[58]. SLC2A12 is a glucose transporter that is also involved in fat metabolism[59]. *YAP1* is an oncogene that is a main effector of the HIPPO tumor suppressor pathway and apoptosis inhibitor[60], processes that could influence retinal ganglion cell survival in glaucoma. In mice, heterozygous deletion of *Yap1* leads to complex ocular abnormalities, including microphthalmia, corneal fibrosis, anterior segment dysgenesis, and cataract[61].

Several proteins encoded by genes within the identified POAG risk loci are targets of some currently approved drugs. For instance, COL4A1 is targeted by ocriplasmin, a collagen hydrolytic enzyme that is currently used to treat vitreomacular adhesion (adherence of vitreous to retina). This drug can degrade the structural proteins including those located at the vitreoretinal surface[62]. Clinical trials are in progress to evaluate ocriplasmin therapy for several eye conditions including macular degeneration, diabetic macular edema, macular hole, and retinal vein occlusion. Some other drug candidates targeting proteins encoded by POAG-associated loci are also currently under consideration for treating dementia and cardiovascular diseases including acitretin, a retinoid receptor agonist targeting RARB, which has been considered for treatment of AD in ongoing clinical trials. Also, dipyridamole, a 3′,5′-cyclic phosphodiesterase inhibitor, targets PDE7B and current clinical trials are testing therapies based on dipyridamole for diseases such as stroke, coronary heart disease, ischemia reperfusion injury, and internal carotid artery stenosis. Given that our pathway analyses highlighted the involvement of vasculature development and blood vessel morphogenesis in POAG pathogenesis, dipyridamole could be a potential therapy for POAG through modulation of blood flow, which can be defective in POAG[63,64]. Further studies to confirm the functionality of these POAG risk genes in vivo and in vitro may support the suitability of repurposing these drugs as alternative treatments for POAG. Moreover, comprehensive fine-mapping is required to identify the most likely causal genes that can be targeted by currently approved drugs.

This study has several strengths and limitations. The main strength includes identification of risk loci contributing to the development of POAG across ethnic groups, an advance over prior POAG GWAS that have mainly focused on individuals from a single ancestry group. We showed that combining GWAS data across ancestries increases the power of gene mapping for POAG. Another strength is the integration of GWAS, gene expression, and chromatin interaction data to investigate the functional relevance of the identified loci, as well as to identify the most plausible risk genes.

A limitation of this study is that although the majority of the cases were clinically confirmed POAG, our data included >7200 glaucoma cases from the UKBB obtained through self-reports. However, we observed a very high concordance between the GWAS results for clinically validated cases versus self-report. Additionally, the vast majority of our results replicated in self-report data from 23andMe. The second limitation of this study is its relatively low statistical power for the subtype-specific analyses (especially for the NTG subset), limiting the ability of this study to identify subtype-specific loci. Larger NTG GWASs are required to dissect the genetic heterogeneity between POAG subtypes. Third, although where possible each participating study adjusted for the effect of age in their association testing prior to the meta-analysis, in a subset of studies, cases and controls were not matched for age and future studies should fully investigate the effect of the identified risk loci across different age strata, particularly for loci where certain alleles are strongly associated with other age-related conditions. Finally, although we investigated the

functional relevance of the identified risk loci using bioinformatic analyses, we did not confirm their functionality in vitro and in vivo. Further studies to investigate the biological roles of these risk loci with respect to POAG pathogenesis in relevant eye tissues will further shed light on the molecular etiology of POAG.

In conclusion, this study identified a strong cross-ancestry genetic correlation for POAG between Europeans, Asians, and Africans, and identified 127 genome-wide significant loci by combining GWAS results across these ancestries. The cross-ancestry data improved fine-mapping of causal variants. By integrating multiple lines of genetic evidence, we implicate previously unknown biological processes that might contribute to glaucoma pathogenesis including intracellular chloride channels, adipose metabolism and YAP/HIPPO signaling.

## Methods

**Study design and participants.** We obtained 34,179 POAG cases and 349,321 controls including participants of European, Asian, and African descent from 21 independent studies across the world. Number of cases and controls, and distribution of age and sex for each study are summarized in Supplementary Data 1. The phenotype definition and additional details such as genotyping platforms for each study are provided in Supplementary Information. For most of the studies, we restricted glaucoma to POAG based on the ICD9/ICD10 criteria. However, considering that POAG constitutes the majority of glaucoma cases in Europeans[65], we also included 7286 glaucoma self-reports from UK Biobank to replicate findings from the ICD9/ICD10 POAG meta-analysis in Europeans and to maximize the statistical power of the final stage meta-analysis (please see below). Informed consent was obtained from all the participants, and ethics approval was obtained from the ethics committee of all the participating institutions.

We performed a four-stage meta-analysis to combine GWAS data from the participating studies. In the first stage, we conducted a meta-analysis of the POAG GWAS in Europeans (16,677 POAG cases and 199,580 controls). In the second stage, we performed independent meta-analyses of POAG GWAS in Asians (6935 cases and 39,588 controls) and in Africans (3281 cases and 2791 controls) (Supplementary Data 1). As part of the second stage, the Asian and African meta-data, as well as data from a GWAS of 7286 self-report glaucoma cases and 107,362 controls of European descent from UKBB were used to validate the findings from the European Caucasian meta-analysis. The UKBB self-report GWAS was completely independent of the UKBB IC9/ICD10 POAG GWAS; all the UKBB POAG cases and controls from the first stage, as well as their relatives (Pi hat >0.2), were removed from the self-report GWAS dataset. In the third stage, we combined the results from stage 1 and 2 to increase our statistical power to identify POAG risk loci across ancestries. In the fourth stage we replicated the stage 3 findings in a dataset from 23andMe.

To investigate sex-specific loci for POAG, we also conducted a meta-analysis of POAG in males and females separately. For this analysis, we had GWAS data from a subset of the overall POAG meta-analysis, including 10,775 cases and 123,644 controls for males, and 10,977 cases and 144,606 controls for females (Supplementary Data 1). Similarly, to identify risk loci for the HTG and NTG subtypes, we performed a subtype-specific meta-analysis using 3247 NTG cases and 47,997 controls, and 5144 HTG cases and 47,997 controls.

**Quality control (QC) and imputation.** Study-specific QC and imputation details have been provided in Supplementary Information. Overall, SNPs with >5% missing genotypes, minor allele frequency (MAF) < 0.01, and evidence of significant deviations from Hardy–Weinberg equilibrium (HWE) were excluded. In addition, individuals with >5% missing genotypes, one of each pair of related individuals (detected based on a p-hat > 0.2 from identity by descent calculated from autosomal markers), and ancestry outliers from each study (detected based on principal component analysis including study participants and reference samples of known ancestry) were excluded from further analysis (for more details please see Supplementary Information).

Imputation for studies involving participants of European descent was performed in Minimac3 using the Haplotype Reference Consortium (HRC) r1.1 as reference panel through the Michigan Imputation Server[66]. However, for a study of Finnish population (FinnGen study), whole-genome sequence data from 3775 Finnish samples were used as reference panel for better population-specific haplotype matching, which results in a more accurate imputation. For studies involving Asian and African participants, imputation was performed using the 1000 Genomes samples of relevant ancestry. SNPs with MAF > 0.001 and imputation quality scores (INFO or $r^2$) > 0.3 were taken forward for association analysis.

**Association testing.** Association testing was performed assuming an additive genetic model using dosage scores from imputation, adjusting for age, sex, and study-specific principal components as covariates, using software such as

PLINK2[66,67], SNPTEST v2.5.1[68,69], SAIGE v0.36.3[70], EPACTS v3.2.6 (https://genome.sph.umich.edu/wiki/EPACTS), and Rvtests v2.0.3[71]. For studies with a large number of related individuals, mixed-model association testing was performed to account for relatedness between people. For the X chromosome analysis, we used the following approach to allow for dosage compensation: females were coded as 0 (homozygous for non-effect allele), 1 (heterozygous), and 2 (homozygous for effect allele) while males were coded as 0 (no effect allele) and 2 (one effect allele). The covariates were the same as for the association testing for the autosomes except removing sex.

To confirm the validity of combining GWAS results across populations comprising different ancestries, we estimated the genome-wide genetic correlation for POAG between the populations of European, Asian, and African descent participating in this study. For this purpose, we used Popcorn v0.9.9[21], a toolset that provides estimates of genetic correlation while accounting for different genetic effects and LD structure between ancestries. For this analysis, the LD scores for each ancestry population were estimated using the 1000G populations (Europeans, Asians, and Africans), and SNPs were filtered based on the default MAF = 0.05 in Popcorn.

We performed within and between ancestry meta-analyses using a fixed-effects inverse-variance weighting approach in METAL (the version released on 2011-03-25)[72] using SNP effect point estimates and their standard errors. The presence of heterogeneity between SNP effect estimates across studies were investigated using the Cochran's Q test implemented in METAL. To identify multiple independent risk variants within the same locus using GWAS summary statistics obtained from the meta-analysis, we used the Conditional and Joint (COJO) analysis implemented in GCTA v1.26[73]. Q–Q and Manhattan plots were created in R-3.2.2, and regional association plots in LocusZoom v1.4[74].

We used the univariate LD score regression[19] intercept for each study separately as well as for the meta-analyzed results to ensure that the test statistics did not include model or structural biases such as population stratification, cryptic relatedness, and model misspecification. To investigate the genetic correlation between POAG and AD, we used bivariate LD score regression[32] using two large AD GWAS meta-analyses[30,31]. To investigate the genetic correlation between POAG and the other traits, we used bivariate LD score regression through the LD Hub platform (http://ldsc.broadinstitute.org/ldhub/).

The association of the POAG risk loci identified in this study with its major endophenotypes, IOP and VCDR, was investigated using summary statistics from a recent GWAS meta-analysis of IOP ($N = 133,492$)[11] and VCDR ($N = 90,939$)[15].

The variance in the POAG familial risk explained by the loci identified in this study ($N = 127$) was calculated based on $\sum_i 2p_i(1 - p_i)\beta_i^2/\log(\lambda P)$, where $p_i$ and $\beta_i$ refer to the MAF and the magnitude of association of the $i$-th SNP, respectively, and $\log(\lambda P)$ is the familial relative risk obtained from observational studies. The estimates for $p_i$ and $\beta_i$ in this study were obtained from UKBB and European POAG meta-analysis, respectively, and $\log(\lambda P)$ was 9.2 estimated in a previous study[75].

**23andMe replication.** We validated the genome-wide significant risk loci from our cross-ancestry meta-analyses (127 independent SNPs) and subtype analyses (7 independent SNPs) in a subset of 23andMe research participants of European descent comprising 43,254 POAG cases and 1,471,118 controls. POAG cases were defined as those who reported glaucoma, excluding those who reported angle-closure glaucoma or other types of glaucoma. Controls did not report any glaucoma. Association testing was performed using logistic regression assuming an additive genetic model, adjusting for age, sex, top five principal components, and genotyping platform as covariates.

**Cross-ancestry fine-mapping.** We used PAINTOR v3.0[76,77] to perform a cross-ancestry fine-mapping for the 127 risk loci identified in this study. For this analysis, the GWAS summary statistics for 1 Mb either side of the lead risk SNPs were extracted from European (including UKBB self-reports), Asian, and African meta-analyses, separately. To account for different LD patterns between ancestries, we created ancestry-specific LD matrices between SNPs using 1000G phase 3 as a reference panel. We allowed for the presence of two causal SNPs per locus. To investigate any advantage of fine-mapping across ancestries, we compared the posterior probabilities of the prioritized causal SNPs in Europeans separately, as well as across ancestries, without including any annotation data.

**Gene-based and pathway-based tests.** Gene-based and gene-set (pathway) based tests were performed using the approach implemented in MAGMA v1.07b[33]. We performed this analysis for each ancestry separately, and the $P$-values were then combined across ancestries using Fisher's combined probability test. The significance threshold for gene-based test was set to $P < 0.05/20174$, and for pathway-based tests to $P < 0.05/10678$, accounting for the maximum number of independent genes/pathways tested. In addition, MAGMA was used to investigate the enrichment of the expression of the significant risk genes in GETX v6 tissues ($P < 1e-03$ accounting for 53 tissues tested).

To identify loci with effect on POAG risk due to modulation of gene expression, we also used alternative gene-based tests that integrate GWAS summary statistics with eQTL data throughout the genome (TWAS-based approaches). For this purpose, we used MetaXcan v0.3.3[38], SMR v0.69[37], and FOCUS v0.5[40]. MetaXcan uses GWAS summary statistics to impute the genetic component of gene

expression in different tissues based on a reference eQTL panel. We used the EyeGEx eQTL data from retina[39], as well as 44 GTEX tissues as reference eQTL data for this study. The GWAS input for this analysis included the summary statistics obtained from the cross-ancestry meta-analysis. We set the significance threshold to Bonferroni-corrected $P < 1.5e-07$, accounting for the maximum number of genes tested ($N = 7209$) in 45 tissues. To investigate the probability of each gene causally driving TWAS associations, we used the approach implemented in FOCUS[40], a probabilistic framework that assigns a posterior probability to each gene. We used the gene expression reference weights provided in FOCUS, which combines the expression weights obtained from GTEX tissues with the weights provided in FUSION[78] obtained from several tissues, including adipose, peripheral blood, whole blood, and brain.

SMR uses a Mendelian Randomization framework to identify genes whose expression is likely modulated through the same variants associated with the outcome of interest (POAG). For the SMR analysis, we used the following eQTL data: CAGE eQTL summary data from blood ($N = 2765$) and EyeGEx eQTL data from retina ($N = 406$). The SMR significance threshold was set to $P < 2.9e-06$, accounting for the maximum number of 8516 genes tested in two tissues. A heterogeneity $P > 0.05$ from the HEIDI test implemented in SMR implies that we cannot reject the null hypothesis that a single causal variant is likely to affect both gene expression and POAG risk for these loci.

**Gene expression**. RNA was extracted from the corneal layers, trabecular meshwork, ciliary body, retina, and optic nerve tissues from 21 healthy donor eyes of 21 individuals. After quality control, the RNA was sequenced using Illumina NextSeq® 500 (San Diego, USA). Trimgalore (v0.4.0) was used to trim low-quality bases (Phred score < 28) and reads shorter than 20 bases after trimming were discarded. Data analysis was done with edgeR (version 3.22.5)[79]. Only genes expressed a minimum of 10 times (1.5 counts per million) in at least five dissected tissues were kept. The RNA count libraries were normalized using trimmed mean of $M$-values method[80]. Two-group differential expression analysis was done via negative binomial generalized linear model in edgeR[81]. The RNA expression in ciliary body, trabecular meshwork, and optic nerve head, which are involved in aqueous production, drainage and principal site of glaucoma injury, respectively, was compared to the remaining eye tissues.

**Drug targets**. We used Open Targets[82] to search for drugs currently in use or in clinical trials for treating other ocular or systemic diseases that target the POAG risk genes identified in this study. These drugs can be potentially repurposed as alternative treatments for POAG, owing to in vivo and in vitro confirmation of the functionality of the target genes in the pathogenesis of POAG.

**Bioinformatic functional analyses**. The bioinformatic functional analysis to investigate the functional relevance of the identified risk loci for POAG were performed through the FUMA platform v1.3.5[83] using the following dataset/toolsets: GTEX eQTL v6[84]; Blood eQTL browser[85]; BIOS QTL Browser[86]; BRAINEAC[87]; RegulomeDB v1.1[36]; CADD v1.4[35]; ANNOVAR (the version released on 2016Feb01)[88]; and Hi-C data from 21 tissue/cell types (GEO accession: GSE87112)[89], PsychENCODE[90], Giusti-Rodriguez et al. (2019)[91], and FANTOM5 Human Enhancer Tracks (http://slidebase.binf.ku.dk/human_enhancers/presets).

**Colocalization analysis**. To test whether the POAG loci tag a shared causal variant or haplotype with Alzheimer's disease (AD), given that several loci overlap, we applied the bayesian-based colocalization method, eCAVIAR v2.0.0[29] to all 123 and 66 autosomal POAG significant variants from the cross-ancestry and European subset POAG GWAS meta-analyses, respectively, and the AD GWAS meta-analysis from Kunkle et al.[30,31] and Jansen et al.[31] that lack summary statistics for chromosome X. Loci were defined by identifying the outermost variant on either side of each lead POAG variant corresponding to $r^2 > 0.1$ relative to the POAG variant with an added 50 kb on either side. We used the default parameters of eCAVIAR (http://genetics.cs.ucla.edu/caviar/manual.html), assuming up to two causal variants per locus. We used the European derived samples from 1000 Genome Project Phase 3 as the reference panel to compute the LD matrix between all variant pairs within each locus needed in eCAVIAR, since both AD GWAS meta-analyses consisted of samples from European ancestry. We used a colocalization posterior probability (CLPP) above 0.01 as the significance cutoff as recommended by the tool[29].

**Reporting summary**. Further information on research design is available in the Nature Research Reporting Summary linked to this article.

## Data availability
The GWAS summary statistics generated in this study are available via GWAS Catalog under study accession identifiers GCST90011766, GCST90011767, GCST90011768, GCST90011769, GCST90011770. UK Biobank data, including POAG, VCDR, IOP, RNFL, and GCIPL GWASs are available by request through the UK Biobank Access Management System https://www.ukbiobank.ac.uk/. The GWAS result from 23andMe are available by request from https://www.23andme.com/. Restrictions apply to the

availability of these data (please see https://www.ukbiobank.ac.uk/principles-of-access/ and https://research.23andme.com/dataset-access/), which were used under license for the current study, and so are not publicly available. The GWAS results for Alzheimer's disease that we used for this study are available from https://ctg.cncr.nl/software/summary_statistics, and by request from https://www.niagads.org/datasets/ng00075. The Haplotype Reference Consortium (HRC) r1.1 is accessible by request from https://www.ebi.ac.uk/ega/studies/EGAS00001001710. Data access requests are reviewed by the Data Access Committee at Wellcome Trust Sanger Institute. This resource can be used for imputation without direct access to the raw data through Michigan Imputation server (https://imputationserver.sph.umich.edu/index.html#!) and Sanger Imputation Service (https://imputation.sanger.ac.uk/). We used HRC r1.1 for imputation through the Michigan Imputation server. The 1000 Genomes phase 3 data is available at https://www.internationalgenome.org/. The datasets we used for the functional analyses in this study are available through: GTEX eQTL v6 (https://gtexportal.org/home/), Blood eQTL, BIOS QTL, EyeGEx data (https://www.ncbi.nlm.nih.gov/geo/query/acc.cgi?acc=GSE115828), BRAINEAC, Hi-C data from 21 tissue/cell types under GEO accession GSE87112, PsychENCODE, Giusti-Rodriguez et al. (2019) (https://www.biorxiv.org/content/10.1101/406330v2), and FANTOM5 Human Enhancer Tracks http://slidebase.binf.ku.dk/human_enhancers/presets. These datasets were used through the FUMA platform (https://fuma.ctglab.nl/). The drug target data was obtained through the Open Targets platform (https://genetics.opentargets.org/).

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

## Acknowledgements

This work was conducted using the UK Biobank Resource (application number 25331) and publicly available data from the International Glaucoma Genetics Consortium. This work was also supported by grants from the National Health and Medical Research Council (NHMRC) of Australia (#1107098; 1116360, 1116495, 1023911), the Ophthalmic Research Institute of Australia, the BrightFocus Foundation, UK and Eire Glaucoma Society and Charitable Funds from Royal Liverpool University Hospital, and International Glaucoma Association-Royal College of Ophthalmologists. P.G. is supported by a NHMRC Investigator Grant (#1173390). S.M., J.E.C., K.P.B., and A.W.H. are supported by NHMRC Fellowships. A.P.K. was funded by a Moorfields Eye Charity Career Development Fellowship, a UK Research and Innovation Future Leaders Fellowship and an Alcon Research Institute Young Investigator Award. The EPIC-Norfolk study has received funding from the Medical Research Council (MR/N003284/1, MC-UU_12015/1, and MC_PC_13048) and Cancer Research UK (C864/A14136). The NEIGHBORHOOD consortium is supported by NIH grants P30 EY014104, R01 EY015473, and R01 EY022305. The FinnGen project is funded by two grants from Business Finland (HUS 4685/31/2016 and UH 4386/31/2016) and eleven industry partners (AbbVie Inc, AstraZeneca UK Ltd, Biogen MA Inc, Celgene Corporation, Celgene International II Sàrl, Genentech Inc, Merck Sharp & Dohme Corp, Pfizer Inc., GlaxoSmithKline, Sanofi, Maze Therapeutics Inc., Janssen Biotech Inc). This work was also supported by the National Eye Institute/National Institutes of Health [R01EY018246, and support from the National Institutes of Health Center for Inherited Disease Research to T.L.Y.]; a University of Wisconsin Centennial Scholars Award [to T.L.Y.]; and an unrestricted grant from Research to Prevent Blindness, Inc. to the UW-Madison Department of Ophthalmology and Visual Sciences [to T.L.Y.]. Additional acknowledgments are supplied in the Supplementary Information file.

## Author contributions

P.G., E.J., P.H., A.P.K., S.P., A.W.H., R.P.J.I., H.C., J.N.C.B., P.B., T.L.Y., V.V., A.H.J.T., J.K., R.B.M., K.S.N., R.L., M.S., N.A., A.J.C., R.H., A.P., L.J.C., C.C., E.N.V., G.T., Y.S., M.Y., T.N., J.R., E.B., M.K.L., N.G.M., O.O., K.H., Y.K., M.A., Y.M., P.J.F., P.T.K., J.E.M., N.G.S., P.K., J.H.K., C.C.P., F.P., P.M., A.J.L., A.P., C.v.D., J.H., C.H., A.A., S.O., S.A., S.W., M.R., L.R.P., C.C.K., M.H., C.K., D.A.M., M.K., T.A., J.C., S.M., and J.W. were involved in data collection and contributed to genotyping. P.G., P.H., X.H., J.S.O., A.V.S., R.P.J.I., H.C., A.Q., N.S.J., J.M.R., A.R.H., P.B., A.I., J.K., S.U., S.S.R., M.S., G.T., H.C., X.W., A.A., M.A., and J.H.K were involved in data analysis. P.G., S.M., and J.W. wrote the first draft of the paper. E.J., A.P.K., P.J.F., P.T.K., A.J.L., A.P., M.H., C.C.K., D.A.M., M.K., T.A., J.C., S.M., and J.W. designed the study and obtained the funding.

## Competing interests

X.W. and A.A. are employed by and hold stock or stock options in 23andMe, Inc. J.W. is a consultant for Allergan, Editas, Maze, Regenxbio and has received sponsored research support from Aerpio Pharmaceuticals Inc. L.P. is a consultant for Eyenovia, Bausch + Lomb, Verily, and Nicox. T.L.Y. serves as a consultant to Aerpio Pharmaceuticals, Inc. A.P.K. is a consultant to Aerie, Allergan, Google Health, Novartis, Reichert, Santen and Thea. All remaining authors declare no competing interests.

## Additional information

[1]QIMR Berghofer Medical Research Institute, Brisbane, QLD, Australia. [2]Division of Research, Kaiser Permanente Northern California (KPNC), Oakland, CA, USA. [3]Twin Research and Genetic Epidemiology, King's College London, London, UK. [4]NIHR Biomedical Research Centre, Moorfields Eye Hospital NHS Foundation Trust and UCL Institute of Ophthalmology, London, UK. [5]Department of Public Health and Primary Care, Institute of Public Health, University of Cambridge School of Clinical Medicine, Cambridge, UK. [6]Geisinger Research, Biomedical and Translational Informatics Institute, Danville, PA, USA. [7]Menzies Institute for Medical Research, University of Tasmania, Hobart, TAS, Australia. [8]Centre for Eye Research Australia, University of Melbourne, Melbourne, VIC, Australia. [9]Department of Ophthalmology, Harvard Medical School, Boston, MA, USA. [10]Department of Population and Quantitative Health Sciences, Case Western Reserve University School of Medicine, Cleveland, OH, USA. [11]Department of Ophthalmology, Flinders University, Bedford Park, SA, Australia. [12]Geisinger Research, Biomedical and Translational Informatics Institute, Rockville, MD, USA. [13]Cleveland Institute for Computational Biology, Case Western Reserve University School of Medicine, Cleveland, OH, USA. [14]Depatment of Ophthalmology, Erasmus MC, Rotterdam, The Netherlands. [15]Department of Epidemiology, Erasmus MC, Rotterdam, The Netherlands. [16]The Rotterdam Eye Hospital, Rotterdam, The Netherlands. [17]Department of Clinical Genetics, Erasmus MC, Rotterdam, The Netherlands. [18]Department of Ophthalmology and Visual Sciences, University of Wisconsin-Madison, Madison, WI, USA. [19]Medical Research Council Human Genetics Unit, Institute of Genetics and Molecular Medicine, University of Edinburgh, Edinburgh, UK. [20]Institute for Molecular Medicine Finland, HiLIFE, University of Helsinki, Helsinki, Finland. [21]Broad Institute of the Massachusetts Institute of Technology and Harvard University, Cambridge, MA, USA. [22]Analytic and Translational Genetics Unit, Massachusetts General Hospital and Harvard Medical School, Boston, MA, USA. [23]Institute of Human Genetics, Universitätsklinikum Erlangen, Friedrich-Alexander-Universität, Erlangen-Nürnberg, Erlangen, Germany. [24]Department of Ophthalmology, KPNC, Redwood City, CA, USA. [25]Department of Ophthalmology, School of Medicine, University of California San Francisco (UCSF), San Francisco, CA, USA. [26]Department of Ophthalmology, Kings College London, London, United Kingdom.

[27]Institute of Ophthalmology, University College London, London, UK. [28]University Hospital Southampton NHS Foundation Trust, Southampton, UK. [29]Faculty of Medicine, University of Southampton, Southampton, UK. [30]Department of Ophthalmology, Inselspital, University Hospital Bern, University of Bern, Bern, Germany. [31]Department of Ophthalmology, University Medical Center Mainz, Mainz, Germany. [32]Institute of Medical Biostatistics, Epidemiology and Informatics, University Medical Center Mainz, Mainz, Germany. [33]Department of Ophthalmology and Visual Sciences, The Chinese University of Hong Kong, Hong Kong, China. [34]Singapore Eye Research Institute, Singapore National Eye Certre, Singapore, Singapore. [35]Ophthalmology & Visual Sciences Academic Clinical Program, Duke-NUS Medical School, Singapore, Singapore. [36]Department of Ophthalmology, Yong Loo Lin School of Medicine, National University of Singapore, Singapore, Singapore. [37]Duke-National University of Singapore Medical School, Singapore, Republic of Singapore. [38]Tohoku Medical Megabank Organization, Tohoku University, 2-1 Seiryo-machi, Aoba-ku, Sendai, Miyagi, Japan. [39]RIKEN Center for Advanced Intelligence Project, 1-4-1 Nihonbashi, Chuo-ku, Tokyo, Japan. [40]Department of Ophthalmology, Tohoku University Graduate School of Medicine, 1-1, Seiryo-machi, Aoba-ku, Sendai, Miyagi, Japan. [41]Department of Retinal Disease Control, Tohoku University Graduate School of Medicine, 1-1, Seiryo-machi, Aoba-ku, Sendai, Miyagi, Japan. [42]Department of Advanced Ophthalmic Medicine, Tohoku University Graduate School of Medicine, 1-1, Seiryo-machi, Aoba-ku, Sendai, Miyagi, Japan. [43]Department of Ophthalmic Imaging and Information Analytics, Tohoku University Graduate School of Medicine, 1-1, Seiryo-machi, Aoba-ku, Sendai, Miyagi, Japan. [44]European Molecular Biology Laboratory, European Bioinformatics Institute (EMBL-EBI), Wellcome Genome Campus, Hinxton, Cambridge, UK. [45]23 and Me Inc., San Francisco, CA, USA. [46]Department of Ophthalmology, University of Ibadan, Ibadan, Nigeria. [47]Division of Ophthalmology, Department of Neurosciences, University of the Witwatersrand, Johannesburg, South Africa. [48]Unit of Ophthalmology, Department of Surgery, University of Ghana Medical School, Accra, Ghana. [49]Sydney Brenner Institute for Molecular Bioscience, Faculty of Health Sciences, University of the Witwatersrand, Johannesburg, South Africa. [50]Laboratory for Statistical Analysis, RIKEN Center for Integrative Medical Sciences, Yokohama, Japan. [51]Laboratory of Complex Trait Genomics, Department of Computational Biology and Medical Sciences, Graduate School of Frontier Sciences, The University of Tokyo, Tokyo, Japan. [52]Department of Ophthalmology, Graduate School of Medical Sciences, Kyushu University, Fukuoka, Japan. [53]Laboratory for Genotyping Development, RIKEN Center for Integrative Medical Sciences, Yokohama, Japan. [54]National Institute for Health Research (NIHR) Biomedical Research Centre at Moorfields Eye Hospital National Health Service Foundation Trust & UCL Institute of Ophthalmology, London, UK. [55]UCL Institute of Ophthalmology, University College London, London, UK. [56]Cardiff Centre for Vision Sciences, College of Biomedical and Life Sciences, Maindy Road, Cardiff University, Cardiff, UK. [57]Program in Genetic Epidemiology and Statistical Genetics, Harvard T.H. Chan School of Public Health, Boston, MA, USA. [58]Channing Division of Network Medicine, Brigham and Women's Hospital, Harvard Medical School, Boston, MA, USA. [59]Department of Ophthalmology and Visual Sciences, The Chinese University of Hong Kong, Hong Kong, China. [60]Centre for Vision Research, Department of Ophthalmology and Westmead Institute for Medical Research, University of Sydney, Sydney, NSW, Australia. [61]Institute for Molecular Medicine Finland (FIMM), University of Helsinki, Helsinki, Finland. [62]Psychiatric & Neurodevelopmental Genetics Unit, Departments of Psychiatry and Neurology, Massachusetts General Hospital, Boston, MA, USA. [63]Broad Institute of MIT and Harvard, Cambridge, MA, USA. [64]Nuffield Department of Population Health, University of Oxford, Oxford, UK. [65]Department of Ophthalmology, Icahn School of Medicine at Mount Sinai, New York, NY 10029, USA. [66]Department of Ophthalmology, Radboud University Medical Center, Nijmegen, The Netherlands. [67]Institute for Molecular and Clinical Ophthalmology, Basel, Switzerland. [68]Department of Medicine, Duke University, Durham, NC, USA. [69]Department of Ophthalmology, Duke University, Durham, NC, USA. [70]Singapore Eye Research Institute, Singapore, Singapore. [71]Duke-NUS Medical School, Singapore, Singapore. [72]Division of Human Genetics, Genome Institute of Singapore, Singapore, Singapore. [73]Centre for Ophthalmology and Visual Science, University of Western Australia, Lions Eye Institute, Nedlands, WA, Australia. [74]RIKEN Center for Integrative Medical Sciences, Yokohama, Japan. [75]Department of Ophthalmology, Flinders University, Flinders Medical Centre, Bedford Park, SA, Australia. [190]These authors contributed equally: Puya Gharahkhani, Eric Jorgenson, Pirro Hysi, Anthony P. Khawaja, Sarah Pendergrass. [191]These authors jointly supervised this work: Michiaki Kubo, Ching-Yu Cheng, Jamie E. Craig, Stuart MacGregor, Janey L. Wiggs. *Lists of authors and their affiliations appear at the end of the paper. ✉email: Puya.Gharahkhani@qimrberghofer.edu.au

## NEIGHBORHOOD consortium

R. Rand Allingham[76], Murray Brilliant[77], Donald L. Budenz[78], Jessica N. Cooke Bailey[10], John H. Fingert[79], Douglas Gaasterland[80], Teresa Gaasterland[81], Jonathan L. Haines[10], Michael Hauser[68], Robert P. Igo Jr[10], Jae Hee Kang[58], Peter Kraft[57], Richard K. Lee[82], Paul R. Lichter[83], Yutao Liu[84,85], Louis R. Pasquale[65] & Syoko Moroi[83], Jonathan Myers[86], Margaret Pericak-Vance[87], Anthony Realini[88], Doug Rhee[89], Julia E. Richards[83], Robert Ritch[90], Joel S. Schuman[91], William K. Scott[87], Kuldev Singh[92], Arthur J. Sit[93], Douglas Vollrath[94], Robert N. Weinreb[95], Janey L. Wiggs[9,191], Gadi Wollstein[91] & Donald J. Zack[96]

[76]Department of Ophthalmology, Duke University Medical Center, Durham, NC, USA. [77]Center for Human Genetics, Marshfield Clinic Research Foundation, Marshfield, WI, USA. [78]Department of Ophthalmology, University of North Carolina, Chapel Hill, NC, USA. [79]Department of Ophthalmology, University of Iowa, College of Medicine, Iowa City, IA, USA. [80]Eye Doctors of Washington, Chevy Chase, MD, USA. [81]Scripps Genome Center, University of California at San Diego, San Diego, CA, USA. [82]Bascom Palmer Eye Institute, University of Miami Miller School of Medicine, Miami, FL, USA. [83]Department of Ophthalmology and Visual Sciences, University of Michigan, Ann Arbor, MI, USA. [84]Department of Cellular Biology and Anatomy, Georgia Regents University, Augusta, Georgia, USA. [85]James and Jean Culver Vision Discovery Institute, Georgia Regents University, Augusta, GA, USA. [86]Wills Eye Hospital, Philadelphia, PA, USA. [87]Institute for Human Genomics, University of Miami Miller School of Medicine, Miami, FL, USA. [88]Department of Ophthalmology, West Virginia University Eye Institute, Morgantown, WV, USA. [89]Department of Ophthalmology, Case Western Reserve University School of Medicine, Cleveland, OH, USA. [90]Einhorn Clinical Research Center, Department of Ophthalmology, New York Eye and Ear Infirmary of Mount Sinai, New York, NY, USA. [91]Department of Ophthalmology, NYU School of Medicine, New York, NY, USA. [92]Department of Ophthalmology, Stanford University School of Medicine, Palo Alto, CA, USA. [93]Department of Ophthalmology, Mayo Clinic, Rochester, MN, USA. [94]Department of Genetics, Stanford University School of Medicine, Palo Alto, CA, USA. [95]Hamilton Glaucoma Center, Shiley Eye Institute, University of California, San Diego, CA, USA. [96]Wilmer Eye Institute, Johns Hopkins University Hospital, Baltimore, MD, USA.

## ANZRAG consortium

Shiwani Sharma[75], Sarah Martin[75], Tiger Zhou[75], Emmanuelle Souzeau[75], John Landers[75], Jude T. Fitzgerald[75], Richard A. Mills[75], Jamie Craig[75], Kathryn Burdon[97], Stuart L. Graham[98], Robert J. Casson[99], Ivan Goldberg[100], Andrew J. White[60], Paul R. Healey[60], David A. Mackey[73] & Alex W. Hewitt[7]

[97]School of Medicine, Menzies Research Institute Tasmania, University of Tasmania, Hobart, TAS, Australia. [98]Ophthalmology and Vision Science, Macquarie University, Sydney, NSW, Australia. [99]South Australian Institute of Ophthalmology, University of Adelaide, Adelaide, SA, Australia. [100]Department of Ophthalmology, University of Sydney, Sydney Eye Hospital, Sydney, NSW, Australia.

## Biobank Japan project

Masaki Shiono[101], Kazuo Misumi[101], Reiji Kaieda[101], Hiromasa Harada[101], Shiro Minami[102], Mitsuru Emi[102], Naoya Emoto[102], Hiroyuki Daida[103], Katsumi Miyauchi[103], Akira Murakami[103], Satoshi Asai[104], Mitsuhiko Moriyama[104], Yasuo Takahashi[104], Tomoaki Fujioka[105], Wataru Obara[105], Seijiro Mori[106], Hideki Ito[106], Satoshi Nagayama[107], Yoshio Miki[107], Akihide Masumoto[108], Akira Yamada[108], Yasuko Nishizawa[109], Ken Kodama[109], Hiromu Kutsumi[110], Yoshihisa Sugimoto[110], Yukihiro Koretsune[111], Hideo Kusuoka[111], Hideki Yanaiag[112], Akiko Nagai[113], Makoto Hirata[114], Yoichiro Kamatani[50], Kaori Muto[115], Koichi Matsuda[116,117], Yutaka Kiyohara[118], Toshiharu Ninomiya[119], Akiko Tamakoshi[120], Zentaro Yamagata[121], Taisei Mushiroda[122], Yoshinori Murakami[123], Koichiro Yuji[124], Yoichi Furukawa[125], Hitoshi Zembutsu[116,126], Toshihiro Tanaka[127,128,129], Yozo Ohnishi[127,130], Yusuke Nakamura[116,131]Michiaki Kubo[74,191]

[101]Tokushukai Hospitals, Okinawa, Japan. [102]Nippon Medical School, Tokyo, Japan. [103]Juntendo University, Tokyo, Japan. [104]Nihon University, Tokyo, Japan. [105]Iwate Medical University, Morioka, Japan. [106]Tokyo Metropolitan Institute of Gerontology, Tokyo, Japan. [107]The Cancer Institute Hospital of JFCR, Tokyo, Japan. [108]Aso Iizuka Hospital, Iizuka, Japan. [109]Osaka Medical Center for Cancer and Cardiovascular Diseases, Osaka, Japan. [110]Shiga University of Medical Science, Otsu, Japan. [111]National Hospital Organization, Osaka National Hospital, Osaka, Japan. [112]Fukujuji Hospital, Tokyo, Japan. [113]Department of Public Policy, Institute of Medical Science, The University of Tokyo, Tokyo, Japan. [114]Laboratory of Genome Technology, Institute of Medical Science, The University of Tokyo, Tokyo, Japan. [115]Department of Public Policy, Institute of Medical Science, The University of Tokyo, Tokyo, Japan. [116]Laboratory of Molecular Medicine, Institute of Medical Science, The University of Tokyo, Tokyo, Japan. [117]Laboratory of Clinical Genome Sequencing, Graduate School of Frontier Sciences, The University of Tokyo, Tokyo, Japan. [118]Hisayama Research Institute for Lifestyle Diseases, Fukuoka, Japan. [119]Department of Epidemiology and Public Health, Graduate School of Medical Sciences, Kyushu University, Fukuoka, Japan. [120]Department of Public Health, Hokkaido University Graduate School of Medicine, Sapporo, Japan. [121]Department of Health Sciences, University of Yamanashi, Yamanashi, Japan. [122]Laboratory for Pharmacogenomics, RIKEN Center for Integrative Medical Sciences, Yokohama, Japan. [123]Division of Molecular Pathology, Institute of Medical Science, The University of Tokyo, Tokyo, Japan. [124]Project Division of International Advanced Medical Research, Institute of Medical Science, The University of Tokyo, Tokyo, Japan. [125]Division of Clinical Genome Research, Institute of Medical Science, The University of Tokyo, Tokyo, Japan. [126]Division of Genetics, National Cancer Center Research Institute, Tokyo, Japan. [127]SNP Research Center, RIKEN Yokohama Institute, Yokohama, Japan. [128]Department of Human Genetics and Disease Diversity, Graduate School of Medical and Dental Sciences, Tokyo Medical and Dental University, Tokyo, Japan. [129]Bioresource Research Center, Tokyo Medical and Dental University, Tokyo, Japan. [130]Shinko Clinic, Medical Corporation Shinkokai, Tokyo, Japan. [131]Section of Hematology/Oncology, Department of Medicine, The University of Chicago, Chicago, USA.

## FinnGen study

Anu Jalanko[61], Jaakko Kaprio[132], Kati Donner[61], Mari Kaunisto[61], Nina Mars[61], Alexander Dada[61], Anastasia Shcherban[61], Andrea Ganna[61], Arto Lehisto[61], Elina Kilpeläinen[61], Georg Brein[61], Ghazal Awaisa[61], Jarmo Harju[61], Kalle Pärn[61], Pietro Della Briotta Parolo[61], Risto Kajanne[61], Susanna Lemmelä[61], Timo P. Sipilä[61], Tuomas Sipilä[61], Ulrike Lyhs[61], Vincent Llorens[61], Teemu Niiranen[133], Kati Kristiansson[134], Lotta Männikkö[134], Manuel González Jiménez[134], Markus Perola[134], Regis Wong[134], Terhi Kilpi[134], Tero Hiekkalinna[134], Elina Järvensivu[134], Essi Kaiharju[134], Hannele Mattsson[134], Markku Laukkanen[134], Päivi Laiho[134], Sini Lähteenmäki[134], Tuuli Sistonen[134], Sirpa Soini[134], Adam Ziemann[135], Anne Lehtonen[135], Apinya Lertratanakul[135], Bob Georgantas[135], Bridget Riley-Gillis[135], Danjuma Quarless[135], Fedik Rahimov[135], Graham Heap[135], Howard Jacob[135], Jeffrey Waring[135], Justin Wade Davis[135], Nizar Smaoui[135], Relja Popovic[135], Sahar Esmaeeli[135], Jeff Waring[135], Athena Matakidou[136], Ben Challis[136], David Close[136], Slavé Petrovski[136], Antti Karlsson[137], Johanna Schleutker[137], Kari Pulkki[137], Petri Virolainen[137], Lila Kallio[137], Arto Mannermaa[138],

Sami Heikkinen[138], Veli-Matti Kosma[138], Chia-Yen Chen[139], Heiko Runz[139], Jimmy Liu[139], Paola Bronson[139], Sally John[139], Sanni Lahdenperä[139], Susan Eaton[139], Wei Zhou[140], Minna Hendolin[141], Outi Tuovila[141], Raimo Pakkanen[141], Joseph Maranville[142], Keith Usiskin[142], Marla Hochfeld[142], Robert Plenge[142], Robert Yang[142], Shameek Biswas[142], Steven Greenberg[142], Eija Laakkonen[143], Juha Kononen[143], Juha Paloneva[143], Urho Kujala[143], Teijo Kuopio[143], Jari Laukkanen[143], Eeva Kangasniemi[144], Kimmo Savinainen[144], Reijo Laaksonen[144], Mikko Arvas[144], Jarmo Ritari[145], Jukka Partanen[144], Kati Hyvärinen[144], Tiina Wahlfors[144], Andrew Peterson[146], Danny Oh[146], Diana Chang[146], Edmond Teng[146], Erich Strauss[146], Geoff Kerchner[146], Hao Chen[146], Hubert Chen[146], Jennifer Schutzman[146], John Michon[146], Julie Hunkapiller[146], Mark McCarthy[146], Natalie Bowers[146], Tim Lu[146], Tushar Bhangale[146], David Pulford[147], Dawn Waterworth[147], Diptee Kulkarni[147], Fanli Xu[147], Jo Betts[147], Jorge Esparza Gordillo[147], Joshua Hoffman[147], Kirsi Auro[147], Linda McCarthy[147], Soumitra Ghosh[147], Meg Ehm[147], Kimmo Pitkänen[148], Tomi Mäkelä[149], Anu Loukola[150], Heikki Joensuu[150], Juha Sinisalo[150], Kari Eklund[150], Lauri Aaltonen[150], Martti Färkkilä[150], Olli Carpen[150], Paula Kauppi[150], Pentti Tienari[150], Terhi Ollila[150], Tiinamaija Tuomi[150], Tuomo Meretoja[150], Anne Pitkäranta[150], Joni Turunen[150], Katariina Hannula-Jouppi[150], Sampsa Pikkarainen[150], Sanna Seitsonen[150], Miika Koskinen[150], Antti Palomäki[151], Juha Rinne[151], Kaj Metsärinne[151], Klaus Elenius[151], Laura Pirilä[151], Leena Koulu[151], Markku Voutilainen[151], Markus Juonala[151], Sirkku Peltonen[151], Vesa Aaltonen[151], Andrey Loboda[152], Anna Podgornaia[152], Aparna Chhibber[152], Audrey Chu[152], Caroline Fox[152], Dorothee Diogo[152], Emily Holzinger[152], John Eicher[152], Padhraig Gormley[152], Vinay Mehta[152], Xulong Wang[152], Johannes Kettunen[153], Katri Pylkäs[153], Marita Kalaoja[153], Minna Karjalainen[153], Reetta Hinttala[153], Riitta Kaarteenaho[153], Seppo Vainio[153], Tuomo Mantere[153], Anne Remes[154], Johanna Huhtakangas[154], Juhani Junttila[154], Kaisa Tasanen[154], Laura Huilaja[154], Marja Luodonpää[154], Nina Hautala[154], Peeter Karihtala[154], Saila Kauppila[154], Terttu Harju[154], Timo Blomster[154], Hilkka Soininen[155], Ilkka Harvima[155], Jussi Pihlajamäki[155], Kai Kaarniranta[155], Margit Pelkonen[155], Markku Laakso[155], Mikko Hiltunen[155], Mikko Kiviniemi[155], Oili Kaipiainen-Seppänen[155], Päivi Auvinen[155], Reetta Kälviäinen[155], Valtteri Julkunen[155], Anders Malarstig[132], Åsa Hedman[132], Catherine Marshall[132], Christopher Whelan[132], Heli Lehtonen[132], Jaakko Parkkinen[132], Kari Linden[132], Kirsi Kalpala[132], Melissa Miller[132], Nan Bing[132], Stefan McDonough[132], Xing Chen[132], Xinli Hu[132], Ying Wu[132], Annika Auranen[156], Airi Jussila[132], Hannele Uusitalo-Järvinen[156], Hannu Kankaanranta[156], Hannu Uusitalo[156], Jukka Peltola[156], Mika Kähönen[156], Pia Isomäki[156], Tarja Laitinen[156], Teea Salmi[156], Anthony Muslin[157], Clarence Wang[157], Clement Chatelain[157], Ethan Xu[157], Franck Auge[157], Kathy Call[157], Kathy Klinger[157], Marika Crohns[157], Matthias Gossel[157], Kimmo Palin[158], Manuel Rivas[159], Harri Siirtola[160] & Javier Gracia Tabuenca[160]

[132]Pfizer, New York, NY, USA. [133]Finnish Institute for Health and Welfare, Helsinki, Finland. [134]THL Biobank, Finnish Institute for Health and Welfare, Helsinki, Finland. [135]Abbvie, Chicago, IL, USA. [136]Astra Zeneca, Cambridge, UK. [137]Auria Biobank, Univ. of Turku, Hospital District of Southwest Finland, Turku, Finland. [138]Biobank of Eastern Finland, University of Eastern Finland, Northern Savo Hospital District, Kuopio, Finland. [139]Biogen, Cambridge, MA, USA. [140]Broad Institute, Cambridge, MA, USA. [141]Business Finland, Helsinki, Finland. [142]Celgene, Summit, NJ, USA. [143]Central Finland Biobank, University of Jyväskylä, Central Finland Health Care District, Jyväskylä, Finland. [144]Finnish Clinical Biobank Tampere, University of Tampere, Pirkanmaa Hospital District, Tampere, Finland. [145]Finnish Red Cross Blood Service, Finnish Hematology Registry and Clinical Biobank, Helsinki, Finland. [146]Genentech, San Francisco, CA, USA. [147]GlaxoSmithKline, Brentford, UK. [148]Helsinki Biobank, Helsinki, Finland. [149]HiLIFE, University of Helsinki, Helsinki, Finland. [150]Hospital District of Helsinki and Uusimaa, Helsinki, Finland. [151]Hospital District of Southwest Finland, Turku, Finland. [152]Merck, Kenilworth, NJ, USA. [153]Northern Finland Biobank Borealis, University of Oulu, Northern Ostrobothnia Hospital District, Oulu, Finland. [154]Northern Ostrobothnia Hospital District, Oulu, Finland. [155]Northern Savo Hospital District, Kuopio, Finland. [156]Pirkanmaa Hospital District, Tampere, Finland. [157]Sanofi, Paris, France. [158]University of Helsinki, Helsinki, Finland. [159]University of Stanford, Stanford, CA, USA. [160]University of Tampere, Tampere, Finland.

## UK Biobank Eye and Vision Consortium

Tariq Aslam[161], Sarah Barman[162], Jenny Barrett[163], Paul Bishop[161], Catey Bunce[164], Roxana Carare[165], Usha Chakravarthy[166], Michelle Chan[167], Valentina Cipriani[168], Alexander Day[167], Parul Desai[167], Bal Dhillon[169], Andrew Dick[170], Cathy Egan[167], Sarah Ennis[165], Paul Foster[54], Marcus Fruttiger[168], John Gallacher[171],

David Garway-Heath[168], Jane Gibson[165], Dan Gore[167], Jeremy Guggenheim[172], Chris Hammond [3], Alison Hardcastle[168], Simon Harding[173], Ruth Hogg[174], Pirro Hysi [3,190] & Pearse A. Keane[168], Peng T. Khaw[54], Anthony Khawaja[4], Gerassimos Lascaratos[167], Andrew J. Lotery[28], Phil Luthert[168], Tom Macgillivray[169], Sarah Mackie[163], Bernadette Mcguinness[174], Gareth Mckay[174], Martin Mckibbin[175], Danny Mitry[169], Tony Moore[168], James Morgan[172], Zaynah Muthy[168], Eoin O'Sullivan[176], Chris Owen[177], Praveen Patel[167], Euan Paterson[174], Tunde Peto[174], Axel Petzold[178], Jugnoo Rahi[179], Alicja Rudnicka[177], Jay Self[165], Sobha Sivaprasad[167], David Steel[167], Irene Stratton[180], Nicholas Strouthidis[167], Cathie Sudlow[169], Caroline Thaung[168], Dhanes Thomas[167], Emanuele Trucco[181], Adnan Tufail[167], Veronique Vitart [19], Stephen Vernon[182], Ananth Viswanathan[167], Cathy Williams[170], Katie Williams[183], Jayne Woodside[174], Max Yates[184], Jennifer Yip[185], Yalin Zheng[173], Robyn Tapp[177], Denize Atan[170], Alexander Doney[181], Naomi allen[171], Thomas Littlejohns[171], Panagiotis Sergouniotis[186] & Graeme Black[186]

[161]Manchester University, Manchester, UK. [162]Kingston University, Kingston, UK. [163]University of Leeds, Leeds, UK. [164]King's College London, London, UK. [165]University of Southampton, Southampton, UK. [166]Queens University, Belfast, UK. [167]Moorfields Eye Hospital, London, UK. [168]UCL Institute of Ophthalmology, London, UK. [169]University of Edinburgh, Edinburgh, UK. [170]University of Bristol, Bristol, UK. [171]University of Oxford, Oxford, UK. [172]Cardiff University, Cardiff, UK. [173]University of Liverpool, Liverpool, UK. [174]Queen's University, Belfast, UK. [175]Leeds Teaching Hospitals NHS Trust, Leeds, UK. [176]King's College Hospital, London, UK. [177]St George's, University of London, London, UK. [178]UCL Institute of Neurology, London, UK. [179]UCL Institute of Child Health, London, UK. [180]Gloucestershire Hospitals NHS Foundation Trust, Gloucester, UK. [181]University of Dundee, Dundee, UK. [182]University Hospital, Nottingham, UK. [183]King's College London, London, UK. [184]University of East Anglia, Norwich, UK. [185]University of Cambridge, Cambridge, UK. [186]The University of Manchester, Manchester, UK.

## GIGA study group

Neema Kanyaro[187], Cyprian Ntomoka[188], Julius J. Massaga[189] & Joyce K. Ikungura[189]

[187]Department of Ophthalmology Muhimbili University of Health and Allies Sciences, Dar es Salaam, Tanzania. [188]Department of Ophthalmology, Comprehensive Community Based Rehabilitation in Tanzania (CCBRT) Hospital, Dar Es Salaam, Tanzania. [189]National Institute for Medical Research (NIMR), Dar es Salaam, Tanzania.

## 23 and Me Research Team

Michelle Agee[45], Stella Aslibekyan[45], Robert K. Bell[45], Katarzyna Bryc[45], Sarah K. Clark[45], Sarah L. Elson[45], Kipper Fletez-Brant[45], Pierre Fontanillas[45], Nicholas A. Furlotte[45], Pooja M. Gandhi[45], Karl Heilbron[45], Barry Hicks[45], David A. Hinds[45], Karen E. Huber[45], Ethan M. Jewett[45], Yunxuan Jiang[45], Aaron Kleinman[45], Keng-Han Lin[45], Nadia K. Litterman[45], Jennifer C. McCreight[45], Matthew H. McIntyre[45], Kimberly F. McManus[45], Joanna L. Mountain[45], Sahar V. Mozaffari[45], Priyanka Nandakumar[45], Elizabeth S. Noblin[45], Carrie A. M. Northover[45], Jared O'Connell[45], Steven J. Pitts[45], G. David Poznik[45], J. Fah Sathirapongsasuti[45], Anjali J. Shastri[45], Janie F. Shelton[45], Suyash Shringarpure[45], Chao Tian[45], Joyce Y. Tung[45], Robert J. Tunney[45], Vladimir Vacic[45] & Amir S. Zare[45]

