## [Peer Review File · Nature Communications]

Reviewers' comments:

Reviewer #1 (Remarks to the Author):

Gharanhkhani et al performed a large multi-ethnic meta-analysis of genome-wide association studies for primary open-angle glaucoma (POAG) identifying several novel loci. They show that effects are mostly consistent between the different ethnicities and interestingly, they report a genetic link between POAG and Alzheimer's disease. Overall the paper is interesting. However, I have some concerns about the claims they make and suggestions on how to improve the paper.

Recently (in January 2020), a paper was published in Nature genetics (NG) which in large part was written by the same group of authors: "Multitrait analysis of glaucoma identifies new risk loci and enables polygenic prediction of disease susceptibility and progression". The NG paper identifies 114 SNPs associating with glaucoma using a multivariate GWAS and the main focus is on a PRS they constructed which predicts glaucoma progression. It is odd that in this paper by Gharanhkhani et al, they do not reference their other recently published paper. They state that they find 127 associating loci out of which 69 are novel. However, some of the loci they claim are novel were already reported in the NG paper, for instance: RSPO1, ABO, KLF5 and ADAMTS8. In the NG paper, they also similarly conduct a gene-based association analysis and a gene set enrichment analysis finding 196 genes and 14 gene sets. The authors need to clearly state what this paper adds on top of their previously published paper, and make sure the loci they claim to be novel are truly novel.

The key finding of this study is the genetic link between Alzheimer's disease and POAG. I have concerns that the reported genome-wide correlation has P-value only 0.049. That is not a strong association and might be observed by chance. I wonder if the authors could replicate the genetic correlation using the self-reported data, or strengthen the result by using the whole European data. Also some mendelian randomization methods could strengthen the claim. For example, it would be interesting to see a plot of the detected POAG variant showing their effect on glaucoma (for the risk increasing allele) vs their effect on AD and if the correlation is significant for the reported variants. All variants that have been reported for AD could also be tested for their effect on glaucoma. The finding of specific loci affecting both trait is interesting, but a locus affecting different traits is often observed without a genome-wide correlation between the traits.

Since the main finding of this study is the genetic link between Alzheimer's disease and POAG, the introduction could have given some information about Alzheimer's. The fact that epidemiological studies have shown that Alzheimer's is more common for those with glaucoma could be mentioned. (For example: <https://www.nature.com/articles/s41598-018-29557-6>, <https://www.ncbi.nlm.nih.gov/pmc/articles/PMC6120118/>). More introduction and discussion on this would highlight the importance of the results.

In general, when exploring correlation between effect sizes, it is better to use the effects of the glaucoma risk increasing allele instead of the minor allele (which I am assuming they do, although they do not state anywhere which allele is used for the effect plots). Using the minor allele can bias the correlation results. Also, when calculating correlation between effect sizes, it is better to weight by allele frequency, for instance weight by $MAF(1-MAF)$.

In the introduction, the authors state that POAG is highly heritable. It would have been interesting to see an estimate of the heritability explained by the SNPs found in the study and compare it to previous estimates. How much are they adding to the missing heritability?

The endophenotypes explored in the paper; IOP, VCDR, GCIP and RNFL could have been better introduced to a general readers.

In the results section where the number of significant loci is summarized it would be good to see a more detailed summary, for example: How many of the novel variants are coding variant and how many are common or low-frequency variants.

In supplementary table 2 and 4, where the associating variants are listed, important information is

missing like allele frequency, which allele is minor allele, imputation info, the coding effects of the variants, how many variant are in high LD ($r^2 > 0.8$) with the reported variant and are any of them coding. Also, no reference is given for variants that are known. For a binary trait the reported effect is usually an OR instead of beta.

They state (lines 87-88) that the one locus that was genome wide significant in the African data was nominally significant in Asians, but the P-value in supplementary table 3 is 0.075, which is not nominally significant (i.e. > 0.05).

The overall correlation between effect sizes of glaucoma and the endophenotypes should be reported. It would be interesting to summarize the effects of the 127 variants on the explored endophenotypes in one heatmap. A heatmap could be more informative than the scatterplots in Figure 4.

In the eQTL analysis, are any of the reported variants the top eQTL variant for the gene they are affecting? If so, that should be highlighted.

When talking about specific variants, their coding effects should be mentioned. The finemapping is unclear. For example, the MAPT variant is highly correlated with many variants (according to the locusplot). Are any of the correlated variants coding variants?

When showing the correlation between effect sizes for different ethnicities, it would be good to state how many showed the same direction of effects. (Lines 69-78)

They state that they find 127 independent loci located $> 1\text{Mb}$ apart (Line 95). It is unclear whether they found any independent secondary variants at some of the reported loci. It is better to refer to N independent variants or SNPs at N loci.

In general when they report an association between a SNP and a phenotype it would be better to also show the OR.

In the figure labels, use $\log(\text{OR})$ instead of beta.

I could not see a reference for the ocular tissue database expression data.

In table 1 and 2, use only 2 or 3 significant figures.

Reviewer #2 (Remarks to the Author):

The authors present a comprehensive GWAS of POAG, describing 127 loci with disease associations. Numerous statistical and bioinformatics tools were used to link these loci with gene function, pathways and functionally relevant tissues. The manuscript has valid results, but the presentation of the results and discussion sections should be improved to aid the reader through the complicated set of steps. The link between POAG and AD should be strengthened or de-emphasized if additional analyses can not be performed. The results are interesting and valuable and enhanced by the extensive bioinformatics investigations that are presented. Additional suggestions for improving the conclusions are listed below.

Major Points

- The first five sections describing the genetic discovery are difficult to follow. The authors could simplify the presentation of these results and present a final summary of the 127 loci indicating the observed effect and analysis from which the result was derived. The endophenotype and sub-type stratified analyses could help organize these loci into logical groups, which would simplify the presentation of these results. Importantly, the sex-specific, endophenotype and sub-type stratified analysis are not discussed.

- The authors make a point that fine-mapping is improved by the including African GWAS, but formal fine-mapping results are not presented or compared across the ancestry-specific analyses and the trans-ancestry analysis. At each locus, a single index variant is presented without consideration of other variants in high LD which could be the causal variant. Specifically, Supp Table 12 could list all variants within each locus that comprise a high confidence credible set.

- In the discussion about overlapping loci between POAG and AD loci, formal co-localization analysis should be used to quantify and justify the conclusion that there is a shared genetic basis of these traits.

- Results vs Discussion:

Some results are described in the discussion for the first time: the BCAS3, OVOL2 loci are not presented in the results. It is mentioned that these loci have different effects between ancestries, but these loci are not labeled in Figure 2, and the effect in different ancestry groups are not presented in the text.

Several sections of the discussion overlaps substantially with the results, repeating main points and the presentation of results (the MAPT section) or are presenting results (CADM2). It is not clear why the association at APBB2 is interesting — if the current African samples overlap the previous analysis, where the association is observed, it is not a surprising result.

The genes implicated as potential drug targets are not listed/described in the results.

The “First HLA association for POAG” section should be in the discussion section of the paper.

The cohort labels in Supp Table 1 need to be improved. What cohort is “Hauser Africans saf”? The cohorts seem to be listed in the Supplementary Note, but the inconsistent labelling made it difficult to go between the table and the Note.

Reviewer #3 (Remarks to the Author):

This large, cross-ancestry meta-analysis of primary open-angle glaucoma identified 69 novel loci of which 55 replicated in an independent dataset. The fantastic analysis is comprehensive and adds substantial knowledge to the field.

Major Concerns

(1) An incredible amount of effort in the manuscript is dedicated to a link with Alzheimer’s disease which does not seem warranted given the results.

(1a) The authors annotate the chromosome 21 locus as APP, but the functional results suggest that MRPL39 may be a better candidate in that region.

(1b) The MAPT region is notoriously challenging because of a large inversion in the area, but the authors do not report on the MAPT haplotypes and only use a simple LD metric to implicate AD despite the lack of association with AD for the top variant in the region. A more careful description of the MAPT haplotype may be helpful, particularly given the large datasets leveraged with potential to examine the region across ancestral groups.

(1c) The authors also report a weak association between APOE and POAG that is present only in the largest meta-analysis. If a candidate analysis of APOE is going to be included, it should fully assess the $\epsilon 2$, $\epsilon 3$, and $\epsilon 4$ haplotypes.

(1d) Moreover, given the modest correlation with APOE that has an incredibly strong association with AD (3-fold increased risk for 1 copy of $\epsilon 4$, 9-14 fold increased risk for 2 copies), it would be helpful to know whether the very weak genetic correlation with AD is merely driven by the strong effect of APOE on AD. Is a genetic correlation still observed if the APOE region is not included in

the LD score regression?

(1e) To some degree, I think the overt focus on AD often detracts from the functional interpretation of variants and may misrepresent that underlying biology. I mention the APP locus above that may be mis-interpreted, but even the APBB2 example seems like a potential mischaracterization: that particular variant (rs59892895) is actually an eQTL for a cholinergic receptor gene, and APP is the focus simply because it is known to play a role in AD. It seems that letting the deep functional annotation data the authors present guide the interpretation and discussion would improve the manuscript.

(2) There is no discussion around age in the manuscript at all, and I can't find age distributions anywhere in the text or the supplements. This is an important factor for an age-related disease, and given the enormous sample size in the present manuscript, a factor that deserves serious attention.

Minor Concerns

(3) How well and on what basis were cases and controls matched? The selection of controls for a phenotype that emerges over the course of aging is of course quite challenging, and some of the limitations and challenges that emerge should at least be mentioned by the authors. For example, some of the studies selected controls >55 and cases >35, others had controls >40, others age matched. It is possible that all of this just washes out, but it likely warrants discussion as to how population differences between cases and controls across centers could confound analysis.

(4) The authors mention that genotyping platform is presented in the supplement, but the data are not present in Supplementary Table 1 or any other supplemental material that I could find. Please update and highlight any contribution (or lack thereof) that genotyping platform had on results.

(5) Sex-stratified and disease sub-group analyses are quite informative. Could the authors also provide interaction p-values for the top sex-stratified signals to help interpret how sex-specific or group-specific the finding might actually be?

(6) The discussion would be greatly improved by focusing in on the fantastic functional data that are presented in the supplement to help the reader understand of the novel loci. Which of the new loci do you think have a high-quality candidate gene that is well supported by multiple lines of evidence, and what does that tell us about disease biology?

(7) A more agnostic genetic correlation analysis incorporating many traits would also be an improvement (rather than picking a single trait based on some of the genetic signals that emerge in the GWAS).

Response to Referees

A large cross-ancestry meta-analysis of genome-wide association studies identifies 44 novel risk loci for primary open-angle glaucoma

Reviewers' comments:

Reviewer #1 (Remarks to the Author):

Recently (in January 2020), a paper was published in Nature genetics (NG) which in large part was written by the same group of authors: “Multitrait analysis of glaucoma identifies new risk loci and enables polygenic prediction of disease susceptibility and progression”. The NG paper identifies 114 SNPs associating with glaucoma using a multivariate GWAS and the main focus is on a PRS they constructed which predicts glaucoma progression. It is odd that in this paper by Gharanhkhani et al, they do not reference their other recently published paper. They state that they find 127 associating loci out of which 69 are novel. However, some of the loci they claim are novel were already reported in the NG paper, for instance: RSP01, ABO, KLF5 and ADAMTS8. In the NG paper, they also similarly conduct a gene-based association analysis and a gene set enrichment analysis finding 196 genes and 14 gene sets. The authors need to clearly state what this paper adds on top of their previously published paper, and make sure the loci they claim to be novel are truly novel.

Response: Our Nat Genet paper (Craig et al., Nat Genet. 2020 Feb; 52(2):160-166.) was not cited because it was still under review when we submitted this manuscript. For the same reason, we did not exclude the loci identified as novel in our Nat Genet paper from the list of novel loci in the current paper. However, we have now updated the manuscript by citing our Nat Genet paper and have updated the list of novel loci. After excluding those already identified in Craig et al., we now have 44 novel loci. We have changed the relevant sections throughout the main text as well as the main and supplementary figures and tables. We have also updated the number of novel genes identified in the gene-based MAGMA analysis after removing genes that were already identified in the gene-based analysis in Craig et al.

The key finding of this study is the genetic link between Alzheimer’s disease and POAG. I have concerns that the reported genome-wide correlation has P-value only 0.049. That is not a strong association and might be observed by chance. I wonder if the authors could replicate the genetic correlation using the self-reported data, or strengthen the result by using the whole European data. Also some mendelian randomization methods could strengthen the claim. For example, it would be interesting to see a plot of the detected POAG variant showing their effect on glaucoma (for the risk increasing allele) vs their effect on AD and if the correlation is significant for the reported variants. All variants that have been reported for AD could also be tested for their effect on glaucoma. The finding of specific loci affecting both trait is interesting, but a locus affecting different traits is often observed without a genome-wide correlation between the traits.

Response: We have further investigated the genetic link between POAG and AD, and have performed the following analyses for POAG and AD: 1) estimated genome-wide genetic correlation using LD score regression approach¹, 2) estimated correlation of effect sizes of genome-wide significant loci, and 3) investigated the overlap of genetic loci (co-localization) between POAG and AD using the approach implemented in eCAVIAR².

For these purposes, we used the GWAS summary statistics from Kunkle et al., Nat Genet 2019, AD GWAS³ in addition to Jansen et al., Nat Genet 2019, AD GWAS⁴ which was used in our original submission.

LD score genetic correlation:

Using the Jansen AD GWAS, we showed a significant genetic correlation ($rg=0.14$, $P=0.049$) between POAG in Europeans (excluding UKBB self-reports) and AD, as originally presented in the manuscript. The results were consistent (but slightly less significant) when UKBB glaucoma self-reports were added (please see the table below). We also used the Kunkle AD GWAS to replicate the genetic correlation between POAG and AD. Although the CIs overlap with the estimates obtained from the Jansen AD GWAS, the genetic correlation between POAG and AD is non-significant ($rg=0.026$, $P=0.7$) using the Kunkle GWAS. We used the European data only for LD score regression, because the LD pattern that the LD score regression is based on, can differ between different ancestries.

Trait 1	Trait 2	rg	se	P
POAG	AD Jansen	0.1448	0.0739	0.049
POAG + UKBB glaucoma self-reports	AD Jansen	0.1009	0.0564	0.0738
POAG	AD Kunkle	0.026	0.0692	0.7077
POAG + UKBB glaucoma self-reports	AD Kunkle	-0.0237	0.0608	0.6959

We also investigated the genetic correlation using SNPs for POAG endophenotypes (IOP and VCDR). For this purpose, we did a conditional analysis of 1) POAG corrected for IOP or 2) POAG corrected for VCDR, based on GWAS summary statistics for these traits, using the mtCOJO approach implemented in GCTA⁵. In addition, we used IOP and VCDR GWASs from UKBB + IGGC meta-analysis⁶ to investigate their correlation with AD. The results are summarized in the table below. Overall, the results are not significant, except for the genetic correlation between IOP and AD using the Kunkle GWAS ($rg=0.1435$, $P=0.0182$). This association was no longer significant after correction for the multiple testing performed.

Trait 1	Trait 2	rg	se	P
POAG corrected for IOP	AD Jansen	0.1178	0.0966	0.2225
POAG corrected for VCDR	AD Jansen	0.1564	0.0844	0.0638

IOP (UKBB + IGGC)	AD Jansen	0.0869	0.055	0.1138
VCDR (UKBB + IGGC)	AD Jansen	0.0234	0.0523	0.655
POAG corrected for IOP	AD Kunkle	-0.0919	0.0909	0.3118
POAG corrected for VCDR	AD Kunkle	0.0488	0.0754	0.5176
IOP (UKBB + IGGC)	AD Kunkle	0.1435	0.0608	0.0182
VCDR (UKBB + IGGC)	AD Kunkle	-0.0303	0.0546	0.5785

Correlation of effect sizes:

We investigated the correlation of effect sizes of the POAG risk loci (from the cross-ancestry GWAS) with AD. We used the Kunkle AD GWAS for this purpose, as UKBB samples are overlapping between Jansen AD GWAS and our POAG meta-analysis, which may result in biased estimates of correlation (please note that this sample overlap does not affect the LDSC genetic correlation results as the LDSC approach explicitly accounts for presence of sample overlap).

Although there was a suggestive correlation of effect estimates between POAG and AD, it was not significant ($r=0.142$, $P=0.126$; see plot A below).

There was a significant correlation between IOP and AD ($r=0.452$, $P=0.000997$). However, the results were mainly driven by *ANGPT1*, *TXNRD2*, *SVEP*, *TFAP2B-PKHD1*, *MIR548F3*, and *FNDC3B* loci (top right corner of plot B below). Although this is an interesting trend, most SNPs are non-significant for AD.

For VCDR vs AD, there was no significant correlation ($r=0.245$, $P=0.313$, plot C below).

A) POAG vs AD

B) IOP vs AD

C) VCDR vs AD

Co-localization of risk loci- eCAVIAR

To assess colocalization for the POAG risk loci, we applied the Bayesian-based colocalization method, eCAVIAR², to the cross-ancestry and European POAG GWAS meta-analyses and the Kunkle AD GWAS data for 21,982 cases and 41,944 controls of European descent. We found no evidence for sharing of causal variants in the 123 (cross-ancestry) or 66 (European) autosomal

POAG loci (Colocalization Posterior Probability (CLPP) < 0.01; Supplementary Table 7). With the Jansen AD GWAS meta-analysis of 71,880 cases and 383,378 controls, there was weak support for colocalization at 6 loci (CLPP=0.01-0.14; 4 loci from the cross-ancestry and 3 from the European POAG meta-analysis with one overlapping locus; see Supplementary Table 8), though none of these POAG loci reached genome-wide significance in the AD GWAS (AD variant P-values on the order of 10⁻⁴ to 0.05). We note that the colocalization results with this larger AD GWAS meta-analysis might be slightly inflated due to the large overlap of UK biobank samples between the POAG and AD meta-analyses. We have added these results into a new paragraph in the main text as well as into Supplementary Tables 7 and 8).

Overall, although the results above suggest the possibility of a link between glaucoma and AD, there is not strong evidence to support this. In light of these new results, as described above, we added a paragraph explaining that overall there is no clear evidence for the POAG risk loci identified in this study affecting AD. We also removed all the previous sections highlighting a genetic link between AD and POAG throughout the manuscript including the title.

Since the main finding of this study is the genetic link between Alzheimer's disease and POAG, the introduction could have given some information about Alzheimer's. The fact that epidemiological studies have shown that Alzheimer's is more common for those with glaucoma could be mentioned. (For example: <https://www.nature.com/articles/s41598-018->

29557-6, <https://www.ncbi.nlm.nih.gov/pmc/articles/PMC6120118/>). More introduction and discussion on this would highlight the importance of the results.

Response: As discussed above, we have now greatly reduced the emphasis on the POAG-AD link.

In general, when exploring correlation between effect sizes, it is better to use the effects of the glaucoma risk increasing allele instead of the minor allele (which I am assuming they do, although they do not state anywhere which allele is used for the effect plots). Using the minor allele can bias the correlation results. Also, when calculating correlation between effect sizes, it is better to weight by allele frequency, for instance weight by $MAF(1-MAF)$.

Response: We thank the reviewer for this comment. As suggested by this reviewer, we estimated the correlation between effect sizes of SNPs for glaucoma risk increasing alleles. We also re-plotted all the figures that illustrate correlation of effect estimates, accordingly. For weighting, as we do not have an accurate estimate of MAF for SNPs in the meta-analysis (MAF is different for each contributing study), to incorporate estimation errors, we used inverse of variance of effect estimates for weighting. This will indirectly incorporate MAF, as SNPs with lower MAF usually have larger SEs. In general, the estimates of correlation decrease to some extent (as expected) after using glaucoma risk increasing alleles. However, it does not change the results and their interpretation significantly.

We have now updated the results section as follows: “There was moderately high cross-ancestry concordance both for genome-wide significant loci and across the genome. For the genome-wide significant SNPs, the European SNP effects were correlated (Pearson correlation coefficient (r)=0.68 [95% confidence intervals (CIs) 0.38-0.97] and r =0.44 [95% CIs 0.20-0.69]) with Asian and African ancestries, respectively (Supplementary Fig 2B and 2C). Of the 68 SNPs available in the Asian meta-analysis, 60 (88%) showed the same direction of effect as European Caucasians, and of the 66 SNPs available in the African meta-analysis, 55 (83%) showed the same direction as European Caucasians. The genetic correlation across the genome estimated using the approach implemented in Popcorn²¹ was even higher: r =0.85 (95% CIs 0.70-1.00) for European-Asian and r =0.75 (95% CIs -0.93 to 2.43) for European-African. Although the concordance amongst the top SNPs was clear for the European-African comparison, larger sample sizes will be required to narrow the CIs on the European-African genome-wide correlation estimate.”

In the introduction, the authors state that POAG is highly heritable. It would have been interesting to see an estimate of the heritability explained by the SNPs found in the study and compare it to previous estimates. How much are they adding to the missing heritability?

Response: The genome-wide significant loci in this study (N=127) collectively explain 9.4% of the POAG familial risk. The previously known loci explain 7.5% of the familial risk while novel loci explain 1.9%. This is expected because the previously known loci have larger effect sizes, and hence more novel loci with smaller effects are required to explain a variance similar to that explained by the previous known loci. We added this information to the manuscript:

Methods: “The variance in the POAG familial risk explained by the loci identified in this study (N=127) was calculated based on $\sum_i 2p_i(1 - p_i)\beta_i^2 / \log(\lambda P)$ where p_i and β_i refer to the MAF and the magnitude of association of the i -th SNP, respectively, and $\log(\lambda P)$ is the familial relative risk obtained from observational studies. The estimates for p_i and β_i in this study were obtained

from UKBB and European POAG meta-analysis, respectively, and $\log(\lambda P)$ was 9.2 estimated in a previous study⁷⁴.”

The endophenotypes explored in the paper; IOP, VCDR, GCIPL and RNFL could have been better introduced to general readers.

Response: We added the following paragraphs to the results section to better introduce these endophenotypes:

“Several highly heritable endophenotypes are related to POAG risk including IOP, structural variation of the optic nerve characterized as vertical cup-to-disc ratio (VCDR) and variation in thickness of the retina cell layers including the retinal nerve fiber layer (RNFL) and the ganglion cell inner plexiform layer (GCIPL)²⁴.”

“Based on IOP levels, POAG can be classified into two major subtypes: high tension glaucoma (HTG) in which IOP is increased (>21 mm Hg), and normal tension glaucoma (NTG) in which IOP remains within the normal range.”

In the results section where the number of significant loci is summarized it would be good to see a more detailed summary, for example: How many of the novel variants are coding variant and how many are common or low-frequency variants.

Response: We have added two tables to summarize the functional annotation and fine mapping of the risk loci identified in this study (Table 1 and Supplementary Table 14). These tables also show whether the lead GWAS variant is coding or in high LD ($r^2 > 0.8$) with a coding variant. For common/rare variants, we have added 1000G allele frequency of SNPs in EURs, ASNs, and AFRs to the Supplementary Table 4. To summarize these, we included the following sentences in the main text:

“Overall, three lead SNPs are protein-altering variants and 12 lead SNP are in high LD ($r^2 > 0.8$) with a protein-altering variant (Supplementary Table 14), suggesting pathogenic effects through protein-coding roles of these variants (e.g. rs61751937 a missense variant in *SVEPI*).” We also added: “All the lead SNPs have MAF > 0.01 in Europeans, except for two variants: rs74315329 (MAF = 0.0026 in 1000G Europeans) a well-known nonsense variant in *MYOC*^{22,23}, and rs190157577 (MAF = 0.0013 in 1000G Europeans) a novel intronic variant in *LINC02141/LOC105371299*.”

In supplementary table 2 and 4, where the associating variants are listed, important information is missing like allele frequency, which allele is minor allele, imputation info, the coding effects of the variants, how many variant are in high LD ($r^2 > 0.8$) with the reported variant and are any of them coding. Also, no reference is given for variants that are known. For a binary trait the reported effect is usually an OR instead of beta.

Response: We made the following changes: 1) Allele frequencies of effect alleles in EURs, ASNs, and AFRs were added to the Supplementary Table 4. 2) Information about functional annotation (e.g. coding variants) are now summarized in Table 1 and Supplementary Table 14. 3) We have added ORs to all the tables that report associations with POAG. 4) We added a reference to the

known loci in the Supplementary Table 4. 5). Imputation quality scores are different for each participating site and providing an average of scores would not be a suitable parameter to show the overall imputation quality of each SNP, however the imputations scores were included as a quality control step as described in each dataset in the supplementary notes.

They state (lines 87-88) that the one locus that was genome wide significant in the African data was nominally significant in Asians, but the P-value in supplementary table 3 is 0.075, which is not nominally significant (i.e. >0.05).

Response: We have updated the sentence as follows: “This locus has not been previously reported for POAG, and in this study, was not associated with POAG in Europeans (P=0.315) and Asians (P=0.075) (Supplementary Table 3).”

The overall correlation between effect sizes of glaucoma and the endophenotypes should be reported. It would be interesting to summarize the effects of the 127 variants on the explored endophenotypes in one heatmap. A heatmap could be more informative than the scatterplots in Figure 4.

Response: The overall correlation of effect sizes between all POAG risk loci and IOP was 0.53 and between POAG and VCDR was 0.31. We have added this information to the main text. As suggested by this reviewer, we have also created a heatmap (Supplementary Figure 4). Consistent with the data provided in the main text, the heatmap shows that the POAG risk SNPs can be clustered into three groups: a large IOP associated cluster, a smaller VCDR cluster, and a subset that does not appear to be related to IOP or VCDR.

We added the following sentences to the main text: “The overall correlation of effect sizes between all POAG risk loci and IOP was 0.53, and between POAG and VCDR was 0.31, in line with previously published genetic correlation estimates²⁵. To better visualize clustering of the POAG SNPs based on their effect on IOP/VCDR, we created a heatmap by clustering SNPs based on Pearson correlation between effect estimates of SNPs on POAG, IOP, and VCDR (Supplementary Fig 4).”

In the eQTL analysis, are any of the reported variants the top eQTL variant for the gene they are affecting? If so, that should be highlighted.

Response: eQTL data have been provided in Supplementary Table 15. We have also added a table summarizing the functional annotation of the risk loci (Supplementary Table 14), which contains a column to highlight which genes are the most likely causal genes based on the post GWAS approaches we used that incorporate eQTL data (MetaXcan, SMR, and FOCUS).

When talking about specific variants, their coding effects should be mentioned. The finemapping is unclear. For example, the MAPT variant is highly correlated with many variants (according to the locusplot). Are any of the correlated variants coding variants?

Response: As mentioned above, we have now provided Table 1 as well as Supplementary table 14 to summarize the functional annotation of the risk loci including whether any variant is a coding

variant or in high LD with a coding variant. In addition, using PAINOTOR we have now performed cross-ancestry fine mapping and these results have been summarized in Supplementary Table 10. For the *MAPT* example, although the most significant SNP for the *MAPT* region is an intronic SNP located within *MAPT*, the most plausible causal variants identified by PAINOTOR are intronic SNPs in *LINC02210-CRHR1*. The eQTL-based approaches also select *CRHR1* and other genes in this region as more likely causal genes over *MAPT*.

We added the following paragraphs to the main text:

“We used multiple lines of genetic evidence to investigate the functional relevance of the identified risk loci, and to prioritize causal variants and target genes. A summary of these results for the novel loci is provided in Table 1, with additional details presented in Supplementary Table 14. The following paragraphs describe these findings in further detail.”

To highlight some of the interesting findings we added: “Integrating data from several lines of evidence described above, as well as the cross-ancestry fine-mapping and genetic pathways, provided support for specific genes potentially influencing POAG risk particularly *RERE*, *VCAMI*, *ZNF638*, *CLIC5*, *SLC2A12*, *YAP1*, *MXRA5*, and *SMAD6* (Table 1 and Supplementary Table 14). For example, rs3777588, a lead GWAS SNP in this study, is an intronic variant within *CLIC5*. The CADD score for this variant is 16.58, providing support for the pathogenicity of this variant (determined by a CADD score >12.37). The lead SNP is also an eQTL for *CLIC5* in both GTEx and retina, and gene-based analysis by incorporating eQTL data supported the involvement of *CLIC5* in POAG risk. This approach also helped to select best genes near the lead SNPs or to shift the focus from the nearest genes to genes further away. For example, rs12846405, a lead GWAS SNP in this study, is an intergenic variant located between *MXRA5* and *PRKX*. Based on integration of eQTL data and gene-based analysis, *MXRA5* was prioritized as the most likely causal gene in this locus.”

When showing the correlation between effect sizes for different ethnicities, it would be good to state how many showed the same direction of effects. (Lines 69-78)

Response: Compared to European Caucasians, of the 68 SNPs available in the Asian GWAS, 60 (88%) showed the same direction of effect, and of the 66 available in the African GWAS, 55 (83%) showed the same direction. We added this information to the section where we show the correlation of effect sizes between ethnicities.

They state that they find 127 independent loci located >1Mb apart (Line 95). It is unclear whether they found any independent secondary variants at some of the reported loci. It is better to refer to N independent variants or SNPs at N loci.

Response: We did not investigate how many independent variants were available within 1Mb in each locus from the cross-ancestry meta-analysis. This is because the differences in LD patterns between ancestries may preclude the identification of independent loci in a multiethnic GWAS using multiethnic GWAS summary statistics. Hence, we reported loci located >1Mb away from each other as independent loci.⁷

In general when they report an association between a SNP and a phenotype it would be better to also show the OR.

Response: We thank the reviewer for pointing this out. We added ORs to all the tables.

In the figure labels, use log(OR) instead of beta.

Response: We changed beta to log(OR) for binary traits (POAG).

I could not see a reference for the ocular tissue database expression data.

We now have cited a previous publication (MacGregor et al, Nat Genet 2018)⁸ that used this dataset.

In table 1 and 2, use only 2 or 3 significant figures.

Table 1 and 2 have now been moved to Supplementary materials (Supplementary Tables 11 and 12). The statistics were rounded to three decimal places, as suggested by the reviewer.

Reviewer #2 (Remarks to the Author):

The authors present a comprehensive GWAS of POAG, describing 127 loci with disease associations. Numerous statistical and bioinformatics tools were used to link these loci with gene function, pathways and functionally relevant tissues. The manuscript has valid results, but the presentation of the results and discussion sections should be improved to aid the reader through the complicated set of steps. The link between POAG and AD should be strengthened or de-emphasized if additional analyses can not be performed. The results are interesting and valuable and enhanced by the extensive bioinformatics investigations that are presented. Additional suggestions for improving the conclusions are listed below.

Response: We thank the reviewer for these encouraging comments. As described above in the response to Reviewer #1, additional analyses did not significantly strengthen the evidence for a genetic link between POAG and AD. Therefore, we have de-emphasized the results and discussion regarding the potential POAG-AD genetic link.

Major Points

- The first five sections describing the genetic discovery are difficult to follow. The authors could simplify the presentation of these results and present a final summary of the 127 loci indicating the observed effect and analysis from which the result was derived. The endophenotype and sub-type stratified analyses could help organize these loci into logical groups, which would simplify the presentation of these results. Importantly, the sex-specific, endophenotype and sub-type stratified analysis are not discussed.

Response: We thank the reviewer for this useful comment and have revised the sections describing the genetic discovery to improve the clarity of the approach. We have continued to describe the approach according to four stages, as that allows for description of the novel loci found for each ethnicity prior to cross-ancestry meta-analysis. We have also described the sex-specific, endophenotype and subtype analyses in separate sections to provide greater clarity.

Specifically, for endophenotypes (IOP/VCDR), as shown in Fig 4 and supplementary table 5, most of the POAG SNPs are also associated with IOP, or VCDR, with a small subset without a clear effect on IOP or VCDR. We also created a heatmap to better visualize clustering of SNPs based on their effects on IOP and VCDR. We added the following sentence to the results section: “To better visualize clustering of the POAG SNPs based on their effect on IOP/VCDR, we created a heatmap by clustering SNPs based on Pearson correlation between effect estimates of SNPs on POAG, IOP, and VCDR (Supplementary Fig 4).”

For the sex-stratified analyses we observed a very high genetic correlation ($r_g=0.99$, $se=0.06$) between POAG in men versus women, with most of the loci being shared between sexes (except for one locus highlighted in the results section). This similarity does not allow stratifying the risk SNPs based on their sex-specific effects. Similarly, we identified a high genetic correlation ($r_g=0.58$, $se=0.08$) between HTG and NTG subtypes. As highlighted in the results section, except for one locus, all NTG loci were at least nominally associated ($P<0.05$) with HTG (and vice versa). On the other hand, the power of the NTG GWAS is limited compared to HTG, further indicating that stratifying POAG loci based on their HTG/NTG effects requires larger sample sizes. However, we highlighted several loci in the main text, as well as in Supplementary Tables 24 and 25, for which there is a significantly larger effect on one subtype. Accordingly, we added the following sentences to the limitation paragraph in the discussion: “The second limitation of this study is its relatively low statistical power for the subtype-specific analyses (especially for the NTG subset), limiting the ability of this study to identify subtype-specific loci. Larger NTG GWASs are required to dissect the genetic heterogeneity between POAG subtypes.”

- The authors make a point that fine-mapping is improved by the including African GWAS, but formal fine-mapping results are not presented or compared across the ancestry-specific analyses and the trans-ancestry analysis. At each locus, a single index variant is presented without consideration of other variants in high LD which could be the causal variant. Specifically, Supp Table 12 could list all variants within each locus that comprise a high confidence credible set.

Response: We thank the reviewer for raising this issue. We have now performed cross-ancestry fine mapping using the approach implemented in the software PAINTOR. We added the following paragraph to the results section:

“Incorporating GWAS data across European, Asian, and African ancestries allowed us to improve fine mapping of the most likely causal variants. For 10 loci (including novel loci *GJAI/HSF2*, *SEPT7*, and *MXRA5/PRKX*), the posterior probability of finding a causal SNP in Europeans improved after including Asian and African data (improvements from posterior probabilities <0.9 to >0.9 or from <0.8 to >0.8 ; Supplementary Table 10). For eight loci (of which *THRB* and *SMAD6* are novel), although the posterior probability of a SNP being causal in Europeans was high (>0.9 at least for one SNP), there was still a slight improvement after including the other ancestries (Supplementary Table 10). In contrast, the cross-ancestry data made fine mapping worse for three loci where the posterior probabilities in Europeans were >0.8 but declined to <0.8 after incorporating data from the other ancestries. For the rest of the loci, the posterior probabilities did not change significantly after including Asian and African data. Overall, the best causal SNPs in Europeans changed for 52 of 127 loci after including data from the other ancestries (Supplementary Table 10). For the remaining 75 loci, at least one SNP remained the best causal SNP in both fine-

mapping using European data alone as well as across ancestries, and 23 of the 127 lead SNPs identified in the meta-analysis remained the best causal SNP in cross-ancestry fine mapping.”

We also added the following paragraph to the method section:

“We used PAINTOR v3.0^{75,76} to perform a cross-ancestry fine mapping for the 127 risk loci identified in this study. For this analysis, the GWAS summary statistics for 1 Mb either side of the lead risk SNPs were extracted from European (including UKBB self-reports), Asian, and African meta-analyses, separately. To account for different LD patterns between ancestries, we created ancestry-specific LD matrices between SNPs using 1000G phase 3 as a reference panel. We allowed for the presence of two causal SNPs per locus. To investigate any advantage of fine mapping across ancestries, we compared the posterior probabilities of the prioritized causal SNPs in Europeans separately, as well as across ancestries, without including any annotation data.”

- In the discussion about overlapping loci between POAG and AD loci, formal co-localization analysis should be used to quantify and justify the conclusion that there is a shared genetic basis of these traits.

Response: We thank the reviewer for raising this point. To address this concern, as presented in response to a comment of the first reviewer, we conducted co-localization analyses using eCAVIAR. See response to reviewer above for more details.

- Results vs Discussion:

Some results are described in the discussion for the first time: the BCAS3, OVOL2 loci are not presented in the results. It is mentioned that these loci have different effects between ancestries, but these loci are not labeled in Figure 2, and the effect in different ancestry groups are not presented in the text.

Response: The results for *BCAS3* locus are shown in Supplementary Table 2 and 4, and for *OVOL2* in Supplementary Table 2 only. This is because *OVOL2* is one of three loci that were genome-wide significant ($P=2.3e-08$) in Europeans, but not after including the other ancestries ($P=0.003455$). We clarified in the main text that: “All loci identified in the European meta-analysis were also significant at the genome-wide level in the combined ancestry meta-analysis except for three loci, two of which are novel (*OVOL2* and *MICAL3*), and one previously reported (*EGLN3/SPTSSA*).”

The section of the discussion pointed out by the reviewer was mainly to highlight similarities of genetic architecture between ancestries and the benefit of cross-ancestry meta-analyses for POAG. Given that we have now added a cross-ancestry fine mapping approach to the results section, we updated this section of the discussion as follows: “We identified a significant correlation between the POAG effect sizes of genome-wide significant SNPs, as well as all the SNPs throughout the genome, across Europeans, Asians, and Africans. Although previous studies have suggested that the genetic architecture of POAG might differ between Africans and Europeans⁴², we observed a moderate correlation ($r \sim 0.45$) between effect sizes of the POAG risk loci in Europeans and Africans (Supplementary Fig 2C), and the correlation was higher between Europeans and Asians ($r \sim 0.7$). Although the overall correlation is moderately high across ancestries, there are genomic regions where the LD pattern differs by ancestry and our fine-mapping approach showed that incorporating GWAS data across ancestries improved the probability of finding a causal variant

for 18 loci in this study, including known (e.g. *AFAP1* and *RELN* loci) and novel (e.g. *GJA1/HSF2* and *SEPT7*) loci. However, the most probable causal variants in Europeans remained the same for ~60% (75 out of 127) of the risk loci even after incorporating Asian and African GWASs. Overall, due to the relatively lower statistical power of our African studies, the fine-mapping results in this study were not strongly influenced by African GWASs, emphasizing that larger African POAG GWASs are required for better cross-ancestry fine mapping in the future.”

Several sections of the discussion overlap substantially with the results, repeating main points and the presentation of results (the MAPT section) or are presenting results (CADM2). It is not clear why the association at APBB2 is interesting — if the current African samples overlap the previous analysis, where the association is observed, it is not a surprising result.

Response: Thank you for this comment. To address concerns regarding the genetic link between POAG and AD, we have performed several additional analyses. Although suggestive, those analyses did not strongly support a genetic overlap between POAG and AD. Hence, we have now removed the emphasis on the AD genes such as *APBB2*.

The genes implicated as potential drug targets are not listed/described in the results.

Response: These genes are listed in Supplementary Table 21 in a column labeled as “Target genes”. These are the genes nearest to the GWAS lead SNPs. We clarified this in the results section, with a further clarification in the discussion section as follows: “Further studies to confirm the functionality of these POAG risk genes in-vivo and in-vitro may support the suitability of repurposing these drugs as novel treatments for POAG. Moreover, comprehensive fine mapping is required to identify the most likely causal genes that can be targeted by currently approved drugs.”

The “First HLA association for POAG” section should be in the discussion section of the paper.

Response: We moved the HLA association to the discussion section.

The cohort labels in Supp Table 1 need to be improved. What cohort is “Hauser Africans saf”? The cohorts seem to be listed in the Supplementary Note, but the inconsistent labelling made it difficult to go between the table and the Note.

Response: We apologize for the inconsistent labeling. We have now made the labels consistent between Supplementary Table 1 and Supplementary Notes.

Reviewer #3 (Remarks to the Author):

Major Concerns

(1) An incredible amount of effort in the manuscript is dedicated to a link with Alzheimer’s disease which does not seem warranted given the results.

Response: We thank this reviewer for this comment. As also discussed in the responses to comments from reviewers 1 and 2, we have performed additional analyses to examine the evidence for a POAG-AD link (details have been presented above in response to a comment of the Reviewer #1). Although the results are suggestive, the additional analysis did not help to strengthen the evidence for a genetic link and did not identify strong overlap with the novel loci identified in this study. Hence, we decided to de-emphasize the discussion of the POAG-AD link in this manuscript. On balance, we believe the inclusion of multiple ancestries is a major novelty of this work (especially given differences in prevalence across ancestries and recent debate over whether the same genes operate across in different populations) and we have taken the opportunity to use the regained space in the manuscript to strengthen the reporting of the cross-ancestry findings.

(1a) The authors annotate the chromosome 21 locus as *APP*, but the functional results suggest that *MRPL39* may be a better candidate in that region.

Response: We agree with the reviewer that based on eQTL results, *MRPL39* might be the best candidate gene in this region. The assigned genes in Supplementary tables 2 and 4 are the nearest well-characterized non-pseudo genes to the lead GWAS SNPs. Hence, *GABPA/APP* was assigned for this locus. However, we have now added two tables (Table 1 and Supplementary Table 14) that summarize the functional annotation and fine-mapping analyses we performed in this study. These tables now show that *MRPL39* is the most significant eQTL gene for this locus. In addition, a column in Supplementary Table 14 (labelled as “mapped genes by integration of eQTL”) highlights the most plausible causal genes based on the approaches we used that incorporate eQTL data (MetaXcan, SMR, and FOCUS) for each locus. However, for the chromosome 21 locus example, neither *MRPL39* nor *APP* passed the significance threshold to support involvement of either of these genes.

(1b) The *MAPT* region is notoriously challenging because of a large inversion in the area, but the authors do not report on the *MAPT* haplotypes and only use a simple LD metric to implicate AD despite the lack of association with AD for the top variant in the region. A more careful description of the *MAPT* haplotype may be helpful, particularly given the large datasets leveraged with potential to examine the region across ancestral groups.

Response: We have now performed cross-ancestry fine mapping using PAINTOR^{9,10} for all 127 significant loci (Supplementary Table 10). Although the most significant SNP for the *MAPT* region is an intronic SNP located within *MAPT*, the most plausible causal variants identified by PAINTOR are intronic SNPs within *LINC02210-CRHR1*. The eQTL approaches also select *CRHR1* and other genes in this region as more likely causal genes over *MAPT*. Given the above fine-mapping results and considering that we could not strengthen the POAG-AD genetic link using the additional analyses we performed, we removed the detailed discussion on *MAPT* and the other AD genes (e.g. *APP* and *CADM2*).

(1c) The authors also report a weak association between *APOE* and POAG that is present only in the largest meta-analysis. If a candidate analysis of *APOE* is going to be included, it should fully assess the $\epsilon 2$, $\epsilon 3$, and $\epsilon 4$ haplotypes.

Response: We strongly agree with the reviewer on this point. As explained above, we have now de-emphasized the link with AD, and have removed the detailed discussion sections on specific AD genes such as *APOE*.

(1d) Moreover, given the modest correlation with APOE that has an incredibly strong association with AD (3-fold increased risk for 1 copy of ϵ 4, 9-14 fold increased risk for 2 copies), it would be helpful to know whether the very weak genetic correlation with AD is merely driven by the strong effect of APOE on AD. Is a genetic correlation still observed if the APOE region is not included in the LD score regression?

Response: As suggested by this reviewer, we repeated the LD score regression analysis after removing the APOE region (1Mb either side of the APOE alleles). The genetic correlation decreases from 0.14 to 0.10 after the APOE region is removed. Although, for the reasons discussed above, in the revised manuscript we have de-emphasized the POAG-AD genetic link.

(1e) To some degree, I think the overt focus on AD often detracts from the functional interpretation of variants and may misrepresent that underlying biology. I mention the APP locus above that may be mis-interpreted, but even the APBB2 example seems like a potential mischaracterization: that particular variant (rs59892895) is actually an eQTL for a cholinergic receptor gene, and APP is the focus simply because it is known to play a role in AD. It seems that letting the deep functional annotation data the authors present guide the interpretation and discussion would improve the manuscript.

Response: We thank the reviewer for these comments. We have now removed the emphasis on the AD genes such as *APBB2/APP/MAPT* and have shifted our focus to other unique and valuable aspects of our study especially the cross-ancestry fine mapping, as well as the functional annotation findings and have added additional annotation tables to summarize the supportive annotation data for each locus.

(2) There is no discussion around age in the manuscript at all, and I can't find age distributions anywhere in the text or the supplements. This is an important factor for an age-related disease, and given the enormous sample size in the present manuscript, a factor that deserves serious attention.

Response: We apologise for not including these details. Age information has been added in Supplementary Table 1. The age of diagnosis or enrollment for cases and controls was available for most but not all studies contributing data to the meta-analysis. Where available, age was included as a covariate in the logistic regression analysis for each site (Supplementary Table 1, Supplementary notes).

Minor Concerns

(3) How well and on what basis were cases and controls matched? The selection of controls for a phenotype that emerges over the course of aging is of course quite challenging, and some of the limitations and challenges that emerge should at least be mentioned by the authors. For example, some of the studies selected controls >55 and cases >35, others had controls >40, others age matched. It is possible that all of this just washes out, but it likely

warrants discussion as to how population differences between cases and controls across centers could confound analysis.

Response: We agree that selecting cases and controls for an age-related phenotype is challenging and may introduce bias. For this study, cases and controls were selected by each contributing study according to established criteria that included: age, genotyping quality (e.g. missing genotype, call rate, allele frequency, HWE, etc.), lack of evidence for batch effect, genetic principal components, ethnicity, and being unrelated to any of the cases. Any variable significantly different between cases and controls was adjusted for as a covariate in the logistic regression model applied to each contributing study as described in the supplemental notes. The age distribution of cases and controls for each contributing study is presented in Supplementary Table 1 as described above.

(4) The authors mention that genotyping platform is presented in the supplement, but the data are not present in Supplementary Table 1 or any other supplemental material that I could find. Please update and highlight any contribution (or lack thereof) that genotyping platform had on results.

Response: The genotyping platform(s) for each study is described in Supplementary Notes. Any confounding effects of genotyping platforms or batch effects were adjusted for by inclusion of covariates in the logistic regression model used by each contributing study as described in the Supplementary Notes.

(5) Sex-stratified and disease sub-group analyses are quite informative. Could the authors also provide interaction p-values for the top sex-stratified signals to help interpret how sex-specific or group-specific the finding might actually be?

Response: We thank the reviewer for this interesting suggestion. In order to perform interaction analyses, we need individual level data from each participating site. Due to informed consent limitations we do not have individual level data for most subjects.

(6) The discussion would be greatly improved by focusing in on the fantastic functional data that are presented in the supplement to help the reader understand of the novel loci. Which of the new loci do you think have a high-quality candidate gene that is well supported by multiple lines of evidence, and what does that tell us about disease biology?

Response: As suggested by the reviewer, we have now highlighted and summarized the functional work that we performed in this study in Table 1 and Supplementary Table 14. These tables include data from multiple lines of evidence including: 1) whether a variant is a coding variant or in high LD with a coding variant, 2) CADD pathogenicity score for the lead SNPs, 3) information on chromatin interactions and eQTL data, 4) identifying genes within the risk loci that are part of a significant pathway, and 5) genes that are prioritized as potentially causal based on integration of eQTL data. In addition, we included the prioritized causal SNPs based on our cross-ancestry fine mapping in PAINTOR. Finally, we added the following paragraph to the results section as an example of how incorporating multiple lines of evidence could help identify the most likely causal variants and target genes: “Integrating data from several lines of evidence described above, as well as the cross-ancestry fine-mapping and genetic pathways, provided support for specific genes potentially influencing POAG risk particularly *RERE*, *VCAM1*, *ZNF638*, *CLIC5*, *SLC2A12*, *YAP1*,

MXRA5, and *SMAD6* (Table 1 and Supplementary Table 14). For example, rs3777588, a lead GWAS SNP in this study, is an intronic variant within *CLIC5*. The CADD score for this variant is 16.58, providing support for the pathogenicity of this variant (determined by a CADD score >12.37). The lead SNP is also an eQTL for *CLIC5* in both GTEx and retina, and gene-based analysis by incorporating eQTL data supported the involvement of *CLIC5* in POAG risk. This approach also helped to select best genes near the lead SNPs or to shift the focus from the nearest genes to other genes located in further distance. For example, rs12846405, a lead GWAS SNP in this study, is an intergenic variant located between *MXRA5* and *PRKX*. Based on integration of eQTL data and gene-based analysis, *MXRA5* was prioritized as the most likely causal gene in this locus.”

We also added in the discussion section: “Integrating several lines of genetic evidence provided support for specific genes within the novel loci that could influence risk through known and novel processes. *MXRA5* and *SMAD6* are both involved in TGF beta mediated extracellular matrix remodeling^{43,44}, a process known to contribute to POAG risk⁴⁵. Additionally, a *SVEP1* missense allele was associated with POAG risk (rs61751937). *SVEP1* encodes an extracellular matrix protein that is essential for lymphangiogenesis in mice, through interaction with *ANGPT2* (the product of another POAG risk gene identified in this study), and modulation of expression of *TEK* and *FOXC2* in knockout mice⁴⁶. Lymphangiogenesis has an important role in the development of Schlemm’s canal required for outflow of fluid from the eye^{47,48}, and two other genes necessary for lymphangiogenesis and Schlemm’s canal development (*TEK*, *ANGPT1*) cause childhood glaucoma^{49,50}. *VCAM1* is an extracellular matrix cell adhesion molecule involved in angiogenesis and possibly regulation of fluid flow from the eye⁵¹. *RERE* mutations are a cause of neurodevelopmental disorders that can involve the eye⁵², providing further evidence for a role of ocular development in adult glaucoma⁵³. While *RERE* has also been associated with VCDR⁵⁴, this is the first association with POAG.”

“Genes involved in biological processes not previously known to contribute to glaucoma have also been implicated by this study. *CLIC5* encodes a chloride channel that functions in mitochondria⁵⁵ and could have a role in ocular fluid dynamics. *ZNF638* is a zinc finger protein that regulates adipose differentiation⁵⁶ and has been implicated in the genetic regulation of height⁵⁷. *SLC2A12* is a glucose transporter that is also involved in fat metabolism⁵⁸. *YAP1* is an oncogene that is a main effector of the HIPPO tumor suppressor pathway and apoptosis inhibitor⁵⁹, processes that could influence retinal ganglion cell survival in glaucoma. In mice, heterozygous deletion of *Yap1* leads to complex ocular abnormalities, including microphthalmia, corneal fibrosis, anterior segment dysgenesis, and cataract⁶⁰.”

(7) A more agnostic genetic correlation analysis incorporating many traits would also be an improvement (rather than picking a single trait based on some of the genetic signals that emerge in the GWAS).

Response: Thank you for this interesting suggestion. We have now performed genetic correlation analyses with all the traits in LD hub. We added the following paragraph to the Results section: “We next investigated genetic correlation between POAG and a range of other traits using bivariate LDSC³¹ through the LD Hub platform (<http://ldsc.broadinstitute.org/ldhub/>). Only glaucoma, self-report glaucoma, and “Other eye problems” were significantly associated after adjustment for multiple testing for 758 traits ($P < 6.6e-05$; Supplementary Table 9). Some other traits in UKBB

such as myopia (short-sightedness); systolic blood pressure; seeing a psychiatrist for nerves, anxiety, tension or depression; and suffering from nerves, showed some evidence for association at $P < 0.003$ (Supplementary Table 9).”

References:

1. Bulik-Sullivan, B. K. *et al.* LD Score regression distinguishes confounding from polygenicity in genome-wide association studies. *Nat. Genet.* **47**, 291–295 (2015).
2. Hormozdiari, F. *et al.* Colocalization of GWAS and eQTL Signals Detects Target Genes. *Am. J. Hum. Genet.* **99**, 1245–1260 (2016).
3. Kunkle, B. W. *et al.* Genetic meta-analysis of diagnosed Alzheimer’s disease identifies new risk loci and implicates A β , tau, immunity and lipid processing. *Nat. Genet.* **51**, 414–430 (2019).
4. Jansen, I. E. *et al.* Genome-wide meta-analysis identifies new loci and functional pathways influencing Alzheimer’s disease risk. *Nat. Genet.* **51**, 404–413 (2019).
5. Zhu, Z. *et al.* Causal associations between risk factors and common diseases inferred from GWAS summary data. *Nat. Commun.* **9**, 224 (2018).
6. Craig, J. E. *et al.* Multitrait analysis of glaucoma identifies new risk loci and enables polygenic prediction of disease susceptibility and progression. *Nat. Genet.* **52**, 160–166 (2020).
7. Reich, D. E. *et al.* Linkage disequilibrium in the human genome. *Nature* **411**, 199–204 (2001).
8. MacGregor, S. *et al.* Genome-wide association study of intraocular pressure uncovers new pathways to glaucoma. *Nat. Genet.* **50**, 1067–1071 (2018).
9. Kichaev, G. *et al.* Integrating functional data to prioritize causal variants in statistical fine-mapping studies. *PLoS Genet.* **10**, e1004722 (2014).
10. Kichaev, G. & Pasaniuc, B. Leveraging Functional-Annotation Data in Trans-ethnic Fine-Mapping Studies. *Am. J. Hum. Genet.* **97**, 260–271 (2015).

Reviewer #1 (Remarks to the Author):

I think the authors have responded well to the comments. I have no further concerns.

Reviewer #2 (Remarks to the Author):

The paper is much improved in this revision. The results focusing on genes and functional relevance of the loci is compelling and well-written.

I recommend that this is accepted for publication.

The last sentence of the discussion highlights several pathways that may be hypothesis generating, based on the results of the manuscript, but not are not conclusively linked. But this is a very minor point.

Reviewer #3 (Remarks to the Author):

The authors fully addressed the concerns raised by other reviewers and me, particularly the criticisms around the previously reported genetic correlation with AD and the previous focus on known AD risk loci. Although the lack of a clear genetic correlation with AD is disappointing, the updated manuscript is strengthened and more accurately focuses on the most robust findings.

Remaining minor concerns:

(1) The analyses of two critical factors, sex and age, remain somewhat underwhelming. It is understandable that the authors cannot provide comprehensive analyses of all potential sub-analyses given space limitations, but age remains an afterthought. The limitations of age and selection differences across studies warrants mention in the discussion, and the emergence of genomic risk loci across age strata remains an important avenue for future work.

Response to reviewers' comments

Reviewer #2 (Remarks to the Author):

The paper is much improved in this revision. The results focusing on genes and functional relevance of the loci is compelling and well-written.

I recommend that this is accepted for publication.

The last sentence of the discussion highlights several pathways that may be hypothesis generating, based on the results of the manuscript, but not are not conclusively linked. But this is a very minor point.

Response: We thank the reviewer for pointing this out. We agree with this reviewer that the highlighted pathways are not conclusively linked. We now have toned down the last sentence of the discussion by replacing “contributing to glaucoma” with “might contribute to glaucoma”:

“By integrating multiple lines of genetic evidence, we implicate novel previously unknown biological processes that might contribute to glaucoma pathogenesis including intracellular chloride channels, adipose metabolism and YAP/HIPPO signaling.”

Reviewer #3 (Remarks to the Author):

The authors fully addressed the concerns raised by other reviewers and me, particularly the criticisms around the previously reported genetic correlation with AD and the previous focus on known AD risk loci. Although the lack of a clear genetic correlation with AD is disappointing, the updated manuscript is strengthened and more accurately focuses on the most robust findings.

Remaining minor concerns:

(1) The analyses of two critical factors, sex and age, remain somewhat underwhelming. It is understandable that the authors cannot provide comprehensive analyses of all potential sub-analyses given space limitations, but age remains an afterthought. The limitations of age and selection differences across studies warrants mention in the discussion, and the emergence of genomic risk loci across age strata remains an important avenue for future work.

Response: We thank the reviewer for this comment. We now have added the following sentence to the discussion section as a limitation of this study:

“Third, although where possible each participating study adjusted for the effect of age in their association testing prior to the meta-analysis, in a subset of studies, cases and controls were not matched for age and future studies should fully investigate the effect

of the identified risk loci across different age strata, particularly for loci where certain alleles are strongly associated with other age-related conditions.”